# Maxwell's Demon at Work: Efficient Pruning by Leveraging Saturation of Neurons

**Simon Dufort-Labbé**                                    *simon.dufort-labbe@mila.quebec*
*Mila, Université de Montréal*

**Pierluca D'Oro**
*Mila, Université de Montréal*

**Evgenii Nikishin**
*Mila, Université de Montréal*

**Irina Rish**
*Mila, Université de Montréal*

**Razvan Pascanu**
*Google DeepMind*

**Pierre-Luc Bacon**
*Mila, Université de Montréal*

**Aristide Baratin**                                    *baratina@mila.quebec*
*Samsung – SAIL Montreal*
*Mila, Université de Montréal*

**Reviewed on OpenReview:** *https://openreview.net/forum?id=nmBleuFzaN*

## Abstract

When training neural networks, dying neurons—units becoming inactive or saturated—are traditionally seen as harmful. This paper sheds new light on this phenomenon. By exploring the impact of various hyperparameter configurations on dying neurons during training, we gather insights on how to improve upon sparse training approaches to pruning. We introduce **Demon Pruning (DemP)**, a method that controls the proliferation of dead neurons through a combination of noise injection on active units and a one-cycle schedule regularization strategy, dynamically leading to network sparsity. Experiments on CIFAR-10 and ImageNet datasets demonstrate that DemP outperforms existing dense-to-sparse structured pruning methods, achieving better accuracy-sparsity tradeoffs and accelerating training by up to $3.56\times$. These findings provide a novel perspective on dying neurons as a resource for efficient model compression and optimization. The code for our experiments is available here.

## 1 Introduction

In neural network training, dying neurons—those that saturate or become inactive during the learning process—have long been considered undesirable (Maas et al., 2013; Xu et al., 2015), leading to performance degradation or loss of plasticity, especially in non-stationary settings (Lyle et al., 2023; Nikishin et al., 2022; Abbas et al., 2023). Past research has proposed solutions such as alternative activation functions like Leaky ReLU or Swish (Maas et al., 2013; Ramachandran et al., 2018), or methods to reset weights (D'Oro et al., 2022; Dohare et al., 2024). However, the role of dead neurons remains poorly understood.

In this work, we reframe the phenomenon of neuron death and saturation as a **resource** for pruning, allowing us to reduce the size of neural networks dynamically during training. Inspired by Maxwell's Demon, which selectively filters molecules in thermodynamics (Maxwell, 1872), we introduce **Demon Pruning (DemP)**, a sparse training method that (i) promotes neuron death by applying asymmetric Gaussian noise to the weights of active neurons and using a one-cycle regularization schedule, and (ii) iteratively removes dead neurons during training. This approach enables highly structured sparsity with minimal performance loss.

While DemP builds upon simple regularization-based methods (Zhou et al., 2016; Lym et al., 2019; Liu et al., 2017), unlike prior approaches passively inducing sparsity, DemP actively promotes unit saturation during training, yielding progressively sparser subnetworks. Compared to existing methods (Lee et al., 2023), a key feature of DemP is the simplification of the pruning process: by inspecting neuron activations with just a few forward passes, dead units are easily identified and pruned at minimum computational costs.

Despite its simplicity, DemP consistently achieves superior accuracy-sparsity tradeoffs compared to strong baselines in structured dense-to-sparse training, such as EarlyCrop (Rachwan et al., 2022) or SNAP (Verdenius et al., 2020). For instance, when pruning a ResNet-18 on CIFAR-10 or a ResNet-50 on ImageNet, DemP delivers up to a 2.5% improvement in accuracy at 80% sparsity compared to those baselines, while also accelerating training by up to 1.23x on ImageNet. This positions DemP as an efficient method for model compression that balances both accuracy and computational cost. Finally, beyond its strong empirical performance, DemP integrates seamlessly into any training pipeline and can be readily combined with existing pruning techniques.

Our primary contributions are:

1. **Insights into Neuron Saturation:** We explore the mechanisms behind neuron saturation, highlighting the significant roles of stochasticity and key hyperparameters such as learning rate, batch size, and regularization (Section 2 and Appendix B).

2. **A Simple Structured Pruning Method:** We introduce DemP, a dynamic dense-to-sparse training method that promotes neuron death, and prunes them in real-time, by applying decaying regularization and injecting noise early in training (Section 3 and Algorithm 1).

3. **Training Speedup and High Compression:** Extensive experiments on CIFAR-10 and ImageNet show that DemP, despite its simplicity, surpasses strong comparable structured pruning baselines in accuracy-compression tradeoffs, while achieving results comparable to unstructured pruning methods and offering significant training speedup (Section 4 and Appendix G).

## 1.1 Related Work

**Dead Neurons.** It is widely recognized that neurons, especially for networks using ReLU activations, can saturate during training (Agarap, 2018; Trottier et al., 2017; Lu et al., 2019). Continual and reinforcement learning literature linked the accumulation of dead units with plasticity loss (Sokar et al., 2023; Lyle et al., 2023; Abbas et al., 2023; Dohare et al., 2024). Closer to our work, Evci (2018) noted the connection between the dying rate and the learning rate, and derived a pruning technique from it.

Mirzadeh et al. (2024) highlighted the strategic use of ReLU activations to achieve inference efficiency in transformer-based models and proposed an approach leveraging post-activation sparsity to improve inference speed in large language models.

**Structured Pruning.** Pruning reduces the size and complexity of neural networks by removing redundant or less important elements, be they neurons or weights (LeCun et al., 1989). Recent advances such as the Lottery Ticket Hypothesis (LTH) (Frankle & Carbin, 2019) demonstrated the existence of subnetworks trainable to performance comparable to their dense counterpart. While unstructured pruning imposes no constraints on which weights to remove, structured methods aim to eliminate entire structures within a network, such as channels, filters, or layers. This results in smaller, faster models that are compatible with existing hardware accelerators and software libraries (Wen et al., 2016; Li et al., 2017). Structured pruning methods often involve a dense training period followed by sparse fine-tuning (Han et al., 2015). Regularization-based approaches for

sparsification after dense training have been widely explored (Louizos et al., 2018; Liu et al., 2017; Ye et al., 2018; Ding et al., 2018; Wang et al., 2019; 2021). The current SOTA for channel pruning at inference, such as Group Fisher (Liu et al., 2021), CafeNet-R (Su et al., 2021), CHIP (Sui et al., 2021) or AutoSlim (Yu & Huang, 2019), all involve either an extended training period or an increased computational budget, negating the potential benefits of pruning for training speedup.

**Dynamic Pruning:** To address these drawbacks, new methods aim to recover a sparse network under a fixed training budget. Regularization-based methods have been adapted to this setting by leveraging group regularization to influence the training dynamics to yield sparser networks (Zhou et al., 2016; Lym et al., 2019), or remarking that batch normalization offers a natural grouping of units under the scale parameter that can be solely regularized (Liu et al., 2017). The LTH inspired multiple methods seeking the winning ticket without iterative pruning. *Sparse-to-sparse* approaches (Lee et al., 2019; Wang et al., 2020)—pruning weights before training—were adapted to work under structured constraints (Verdenius et al., 2020). Because deep networks suffer increasingly from pruning at initialization (Frankle et al., 2020), *dense-to-sparse* methods perform pruning at a later stage (You et al., 2020; Rachwan et al., 2022).

A recent trend, Dynamic Sparse Training (DST) (Mocanu et al., 2018; Evci et al., 2020), initializes networks sparsely and uses an iteratively updated pruning mask to balance performance and computational efficiency. Initially focusing on unstructured sparsity, DST has recently been adapted for partially-structured sparse networks (Lasby et al., 2024; Yin et al., 2023), yet still not exceeding 50% node sparsity.

**Contextualizing DemP:** Within the landscape of pruning techniques, DemP represents an advancement as a dense-to-sparse end-to-end structured method, offering tangible speedup during both training and inference on GPU hardware. It revisits earlier group regularization-based approaches—monitoring activations similar to Hu et al. (2016) and leveraging increasing regularization as in (Wang et al., 2021) on scale parameters (Liu et al., 2017)—by integrating fresh insights into its design to achieve competitiveness with more recent methods while preserving simplicity. Notably, addressing the phenomenon wherein noisier environments can make saturated regions more attractive (see Lemmas B.2, B.3 and Appendix D.1), DemP introduces asymmetric noise during optimization to further promote sparsity. Sparsity naturally emerges from the learning dynamics with DemP, simplifying the pruning process and eliminating the need for discrete pruning interventions which could disrupt training. Non-contributing neurons can be pruned as they become inactive, without compromising model performance (as demonstrated in Fig. 7). Recent unstructured pruning approaches manually tailor sparsity distribution across layers (Mocanu et al., 2018; Evci et al., 2020), while structured pruning mostly relies on uniform layer-wise pruning (Rachwan et al., 2022). DemP alleviates this by allowing the learning dynamics to determine sparsity distribution across layers.

DemP eliminates the need for meticulous pruning timing (Wang et al., 2021; Rachwan et al., 2022) or heuristic-based pruning schedules (Lee et al., 2023). Acknowledging that early pruning can be detrimental (Frankle et al., 2020; Rachwan et al., 2022), DemP employs easy-to-tune regularization schedules to influence when units die. Additionally, iterative pruning (Verdenius et al., 2020) within a single intervention is unnecessary, as units naturally become inactive during early training with DemP.

**Choice of Baselines:** We highlight and benchmark against strong comparable baselines that use criteria based on gradient flow to evaluate which nodes to prune (Verdenius et al., 2020; Wang et al., 2020; Rachwan et al., 2022). Other methods employing either L0 or L1 regularization on subgroups of parameters to enforce sparsity (Liu et al., 2017; Louizos et al., 2018; You et al., 2019; Lym et al., 2019) are omitted from the benchmark as they are outperformed by Rachwan et al. (2022).

## 2  Saturating Units: An Analysis

In this section, we offer theoretical insights into the occurrence of dead neurons and explore the impact of different training heuristics and hyperparameters. Given a neural network and a set of $n$ training data samples, we denote by $a_j^\ell \in \mathbb{R}^n$ the vector of activations of the $j$th neuron in layer $\ell$ for each training input. We adopt the following definition of a *dead neuron* throughout the paper:

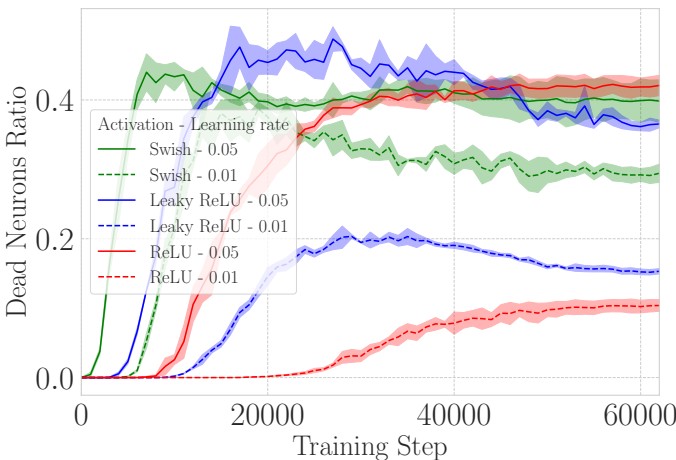

Figure 1: Dead neuron accumulation for a ResNet-18 trained on CIFAR-10 with different activation functions and values of the learning rate. We use a negative slope of $\alpha = 0.05$ for Leaky ReLU and $\beta = 1$ for Swish.

**Definition:** The $j$-th neuron in layer $\ell$ is *inactive* if it consistently outputs zero on the entire training set, i.e. $a_j^\ell = 0$.[1] A neuron that becomes and remains inactive during training is considered as *dead*.[2]

Many modern architectures use activation functions with a saturation region that typically includes 0 at its boundary. In this case, when a neuron becomes inactive during training, its incoming weights also receive zero—or very small[3]—gradients, which makes it difficult for the neuron to recover. In this paper, we mostly work with the Rectified Linear Unit (ReLU) activation function, $\sigma(x) = \max(0, x)$. In this case, the activity of a neuron depends on the sign of the corresponding pre-activation feature.

### 2.1 Neurons Die During Training

We begin with some empirical observations. Applying the above criterion (Footnote 1) with threshold parameter $\epsilon = 0.01$, we monitor the accumulation of dead neurons during training of a Resnet-18 (He et al., 2016) on CIFAR-10 (Krizhevsky et al., 2009) with the Adam optimizer (Kingma & Ba, 2015), with various learning rates and choices of activation functions.

The outcomes are shown in Fig. 1, revealing a notable and sudden increase in the number of inactive neurons early in training. Moreover, a minimal portion of these inactive neurons shows signs of recovery in later stages of training (see Fig. 8 in Appendix C). Overall, this results in a significant fraction of the 3904 neurons/filters in the convolutional layers of the ResNet-18 dying during training, particularly with a high learning rate. We note that this phenomenon is not exclusive to ReLU activations.

**Intuition.** Similar to Maxwell's demon thought experiment (Maxwell, 1872), one can picture an asymmetric behavior of how weights can move across the boundary that delimitates the saturated versus the non-saturated state of a unit, akin to the demon's selective passage of molecules. Neurons, or more specifically their weights, can move freely within the active region, but once they enter the inactive region their movement is impeded, similar to how the demon can trap molecules in a low-energy subspace. In this context, with a random component to their movement, neuron death can be influenced by various factors, e.g., noise from the data, being too close to the border, or taking too large gradient steps. Once the neuron moves into the inactive zone, neurons can only be reactivated if the boundary itself shifts. This asymmetry makes it more likely for neurons to die than to revive.

We formalize this analogy with simple theoretical models in Appendix B. These models are meant to capture the multiplicative (i.e. parameter-dependent) nature of the gradient noise in SGD. Multiplicative noise in

---

[1] In practice, especially for non-ReLU activation functions, we will be using the notion of $\epsilon$-inactivity, defined by the condition $|a_i^\ell| < \epsilon$, for some threshold parameter $\epsilon$.

[2] For convolutional layers we treat channels as neurons (see Appendix A).

[3] e.g., due to regularization.

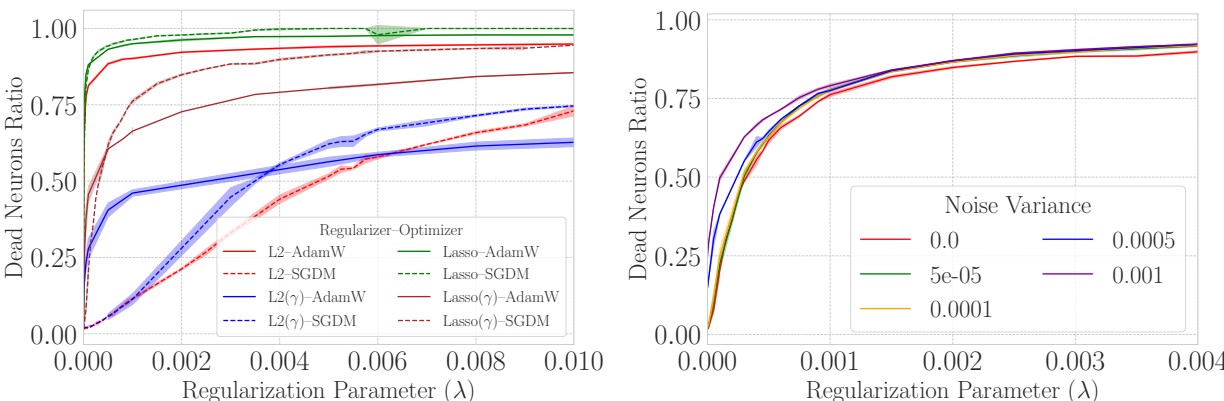

Figure 2: **Left:** Increased regularization during training increases the ratio of dead units, as showcased here for a ResNet-18 trained on CIFAR-10. We use $\cdot(\gamma)$ to denote when regularization is applied solely to the scale parameters of the normalization layers. **Right:** Augmenting training updates with asymmetric Gaussian noise, sampled from $\mathcal{N}(x|0, \sigma^2)$ and applied to weights of live neurons only, also leads to higher levels of dead unit accumulation, also showcased for a ResNet-18 trained on CIFAR-10 with various values of the Lasso$(\gamma)$ regularization parameter.

stochastic processes is known to cause regions with lower noise magnitude to act as attractors (Oksendal, 2010). Essentially, lower noise regions (such as the inactive region of a saturating unit) tend to retain the system due to reduced impact, while higher noise regions push it away. We explicitly demonstrate this phenomenon in the theoretical framework of Appendix B.

## 2.2 Factors Impacting Dying Ratios

**Role of regularization.** The above intuition suggests that maintaining activations close to zero increases the likelihood of neuron death, aligning them closer to the inactive region (the negative domain for ReLU networks). As illustrated in Fig. 2, training a ResNet-18 on CIFAR-10 (with Adam or SGD+momentum) with a higher L2 or Lasso regularization leads to sparser networks caused by an accumulation of dead units. Section 3 elaborates on the approach with DemP, where we opt to apply regularization exclusively on the scale parameters of the normalization layers, following Liu et al. (2017). This tends to better preserve performance and yields significantly improved accuracy-sparsity tradeoffs, as shown in Fig. 5 in Appendix F.

**Role of noise.** Our simple model suggests a pivotal role played by the noise of the training process in the occurrence of dying neurons. To investigate this in a simple setting, we train small MLPs on a subset of 10,000 images of the MNIST dataset in three different noisy regimes: vanilla SGD, pure SGD noise (obtained by isolating the noisy part of the minibatch gradient), and pure Gaussian noise. As observed in Fig. 9 in Appendix D.1, while pure SGD noise training yields dying ratios comparable to SGD, training with pure Gaussian noise is much less prone to dead neuron accumulation. We hypothesize that this difference is due to the asymmetry of SGD noise: since the gradients are 0 for dead neurons, only (the weights of) live neurons are subject to noisy updates. In contrast, the weights of inactive neurons are updated under Gaussian noise training, which increases their probability of recovery. As we spell out in Section 3, DemP injects Gaussian noise exclusively to weights of live neurons during training, which is effective in encouraging neuron death (see Fig. 2 and Fig. 9, middle plot).

We note that other ways to control the SGD noise variance include varying the learning rate and/or the batch size. Fig. 10 (right) in Appendix D.1 illustrates how increasing the learning rate or reducing the batch size indeed increases dying ratios. However, it proved more challenging to balance accuracy and sparsity by adjusting these hyperparameters, as they also significantly affect overall performance. Instead, DemP maintains these parameters at their default or task-optimized values, facilitating its integration into existing training routines.

---

**Algorithm 1** DemP Algorithm

---

**Input:** Learning rate $\eta$, pruning frequency $\tau$, regularization one-cycled schedule $\lambda_t$, noise variance one-cycled schedule $\sigma_t^2$, learnable weights $w$—including normalization scale parameters $\gamma$
**Initialize:** weights $w_0$—including normalization scale parameters $\gamma_0$
**for** $t = 0$ **to** $T$ **do**
    Compute gradient: $\nabla L(w_t)$
    Regularization term: $\nabla R(\gamma_t; \lambda_t)$ affecting only the normalization scale parameters
    Sample asymmetric noise: $\xi_{\text{asym}} \sim \mathcal{N}(0, \sigma_t^2)$
    Update weights:
    $w_{t+1} \leftarrow w_t - \eta(\nabla L(w_t) + \nabla R(\gamma_t; \lambda_t)) + \xi_{\text{asym}}$
    **if** $t\%\tau = 0$ **then**
        Prune dead neurons
    **end if**
**end for**

---

**Optimizer.** We expect the choice of optimizer to affect the final count of dead neurons post-training (e.g., for adaptive optimizers, by altering the effective learning rate per parameter). Notably, we observe a significant discrepancy between using the Adam optimizer (Kingma & Ba, 2015) and SGD with momentum (SGDM) (Fig. 2). As also emphasized by Lyle et al. (2023), we attribute this difference primarily to the specific hyperparameter selection for Adam $(\beta_1, \beta_2, \epsilon)$, which has a substantial impact on neuron death (further discussed in Appendix E). Results from experiments with both optimizers are presented in the next section.

Finally, the total training time and network width can also impact the occurrence of dying neurons and are explored respectively in Appendix D.2 and Appendix D.3.

## 3 Demon Pruning

Leveraging insights from Section 2, we introduce Demon Pruning (DemP), a structured pruning method that dynamically induces neuron saturation during training to achieve high levels of sparsity. DemP achieves this through two primary mechanisms: (1) a dynamic regularization schedule applied to normalization scale parameters and (2) asymmetric noise injection targeting active neurons. The combination of these mechanisms ensures sparse networks without compromising performance. DemP is summarized in Algorithm 1.

### 3.1 Regularization Schedule

Analyzing the pre-activations $(\hat{\mathbf{z}}^{[l]})$ after batch normalization of layer $l$'s output: $(\mathbf{F}_l(\mathbf{W}^{[l]}, \boldsymbol{a}^{[l-1]}))$

$$\hat{\boldsymbol{z}}^{[l]} = \boldsymbol{\gamma}^{[l]} \odot \left( \frac{\mathbf{F}_l(\mathbf{W}^{[l]}, \boldsymbol{a}^{[l-1]}) - \mu^{[l]}}{\sqrt{\sigma^{2[l]} + \epsilon}} \right) + \boldsymbol{\beta}^{[l]} \tag{1}$$

suggests two strategies to push $\hat{\mathbf{z}}^{[l]}$ toward zero: (1) regularizing all networks parameters, (2) regularizing the scale normalization parameters $(\boldsymbol{\gamma})$. We refrain from regularizing the offset parameters $(\boldsymbol{\beta})$, to allow for large negative offsets that can help to get negative pre-activations, leading to dead neurons post ReLU activation. By default, our method employs $\ell_1$-regularization $R(\boldsymbol{\gamma}; \lambda) = \lambda \|\boldsymbol{\gamma}\|_1$ on the scale parameters of normalization layers, as in Liu et al. (2017), with a small modification. Based on the insights of Section 2, we adopt a one-cycled schedule (Smith & Topin, 2018) $\lambda_t$ for the regularization strength. This schedule involves an initial warmup phase with a linear growth to a peak value $\lambda$, followed by a cosine decay to fine-tune sparsity levels while maintaining performance.

Such a one-cycled strategy—originally applied to learning rates—effectively exploits the benefits of both warmup and decay phases (Fig. 4). While regularizing all network parameters is a viable alternative for inducing sparsity, applying regularization exclusively to normalization scale parameters proves more effective and robust, as shown in Fig. 2 and in Section 4.2.

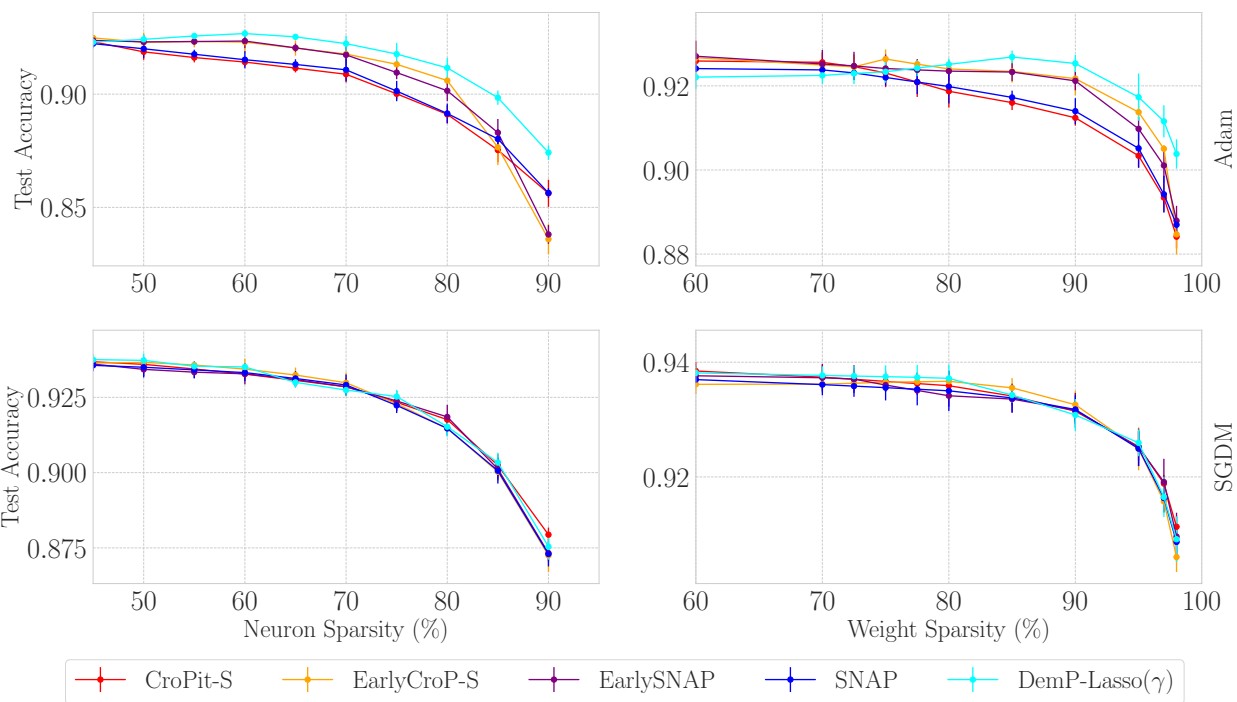

Figure 3: **Top:** For ResNet-18 networks on CIFAR-10 trained with Adam (ReLU), DemP can find sparser solutions while maintaining better performance than other structured approaches. Higher levels of sparsity with DemP are obtained by increasing the peak strength of the added scheduled regularization. **Bottom:** With SGDM, DemP performance is comparable, without significant differences between methods. **Left:** Neural sparsity, structured methods. **Right:** Weight sparsity, structured methods.

### 3.2 Noise Injection

To promote neuron death, DemP injects Gaussian noise selectively into active weights during training. At each iteration, the noise is sampled as $\xi_{\text{asym}} \sim \mathcal{N}(0, \sigma_t^2)$ where the variance $\sigma_t^2$ follows the same one-cycled schedule as the regularization strength. the targeted noise injection enhances sparsity without degrading performance (see Fig. 6). The effectiveness of artificial noise is further validated in Fig. 13 where it demonstrates the ability to sparsify networks even in the absence of regularization.

### 3.3 Dynamic Pruning

DemP leverages targeted regularization and asymmetric noise to progressively accumulate inactive neurons, as described in Section 2. Inactive neurons are identified as those with outputs consistently equal to 0 across the dataset—a criterion relaxed in practice to minibatches (Fig. 12). For Leaky ReLU and GeLU networks, neurons with consistently negative activations are similarly deemed inactive.

Instead of deferring pruning until training concludes, DemP dynamically prunes dead neurons every $t$ iterations (default: 5000). This approach accelerates training by removing pruned neurons from the computation graph, reducing the embedding dimension. Importantly, it allows non-ReLU networks to recover without affecting final performance or sparsity levels (Fig. 7).

DemP thus introduces two new hyperparameters: scheduled regularization on normalization scale parameters and artificial noise. These additions are orthogonal to standard hyperparameters like weight decay, ensuring compatibility with existing training pipelines. By retaining this orthogonality, DemP integrates seamlessly into diverse setups, while leaving room for exploration of additional hyperparameters to achieve greater sparsity. Additional implementation details and ablations are provided in Appendix F.

## 4 Empirical Evaluation

Through extensive empirical evaluation across various benchmarks, we consistently observe that DemP achieves superior performance-sparsity tradeoffs, particularly at high sparsity levels and when combined with Adam. Comparing with SOTA DST methods, such as Yin et al. (2023) and Lasby et al. (2024), that perform restructuration exposes the benefits of DemP for achieving GPU-supported training FLOPs reduction (Table 2).

**Setup:** We focus our experiments on computer vision tasks, which is standard in pruning literature (Gale et al., 2019). We train ResNet-18 and VGG-16 networks on CIFAR-10, and ResNet-50 networks on ImageNet (He et al., 2016; Simonyan & Zisserman, 2015; Krizhevsky et al., 2009; Deng et al., 2009). We follow the training regimes from Evci et al. (2020) for ResNet architectures and use a setting similar to Rachwan et al. (2022) for the VGG to broaden the scope of our experiments. The results for pruning the VGG are reported in Appendix G and training details are provided in Appendix I.

Our method is a structured pruning approach that removes entire neurons during training, transitioning from a dense network to a sparse one. The methods we compare with follow this paradigm. We employ the following structured pruning baselines: Crop-it/EarlyCrop (Rachwan et al., 2022), SNAP (Verdenius et al., 2020) and a modified version using the early pruning strategy from Rachwan et al. (2022) (identified as EarlySNAP). We trained these baselines using the recommended configuration of the original authors; in particular, we did not apply the regularization schedule used in our method.

### 4.1 Results

DemP demonstrates superior performance when paired with Adam optimizer, consistently outperforming all baselines at high sparsity across the board. The margin in test error can reach up to 2.25% when training on CIFAR-10 and ImageNet (Fig. 3 and 15, and Table 1). Notably, the highest sparsification maintaining performance results in a 2.45× faster training on ResNet-18 and 3.56× on VGG-16. The results are more contrasted with SGDM: DemP performs similarly to baselines on ResNet-18 benchmarks (see Fig. 3 and 15), is outperformed at higher sparsities on VGG-16 (see Fig. 14) but slightly outperforms baselines when training

Table 1: Results when pruning a ResNet-50 trained on ImageNet with **Adam** at approximately 80% and 90% weight sparsity. **DemP improves test accuracy by 2.85% and 2.14%** at the respective sparsities. Because structured pruning methods lack precise control over weight sparsity levels., we report the closest numbers obtained to these target values. SNAP and CroPit-S achieve better speedup but do so by reducing the test accuracy below 30%. DemP achieves better speedups compared with methods maintaining accuracy above 60%. Standard deviation ($\pm$) was computed over three seeds.

| | Method | Test accuracy | Neuron sparsity | Weight sparsity | Training time | Training FLOPs | Inference FLOPs |
|---|---|---|---|---|---|---|---|
| | Dense | 74.98% $\pm 0.08$ | - | - | 1.0× | 1.0× (3.15e18) | 1.0× (8.2e9) |
| Structured | SNAP | 28.28% $\pm 0.08$ | 36.9% | 81.4% | 0.51× | 0.32× | 0.32× |
| | | 27.17% $\pm 0.07$ | 56.0% | 90.1% | 0.48× | 0.25× | 0.25× |
| | CroPit-S | 28.34% $\pm 0.52$ | 36.9% | 81.4% | 0.52× | 0.32× | 0.32× |
| | | 27.36% $\pm 0.16$ | 53.2% | 89.9% | 0.47× | 0.27× | 0.27× |
| | EarlySNAP | 68.67% $\pm 0.15$ | 51.70% | 80.37% | 0.95× | 0.63× | 0.63× |
| | | 63.80% $\pm 0.58$ | 66.6% | 90.06% | 0.75× | 0.46× | 0.45× |
| | EarlyCroP-S | 68.26% $\pm 0.31$ | 51.60% | 79.97% | 0.94× | 0.66× | 0.66× |
| | | 64.20% $\pm 0.27$ | 66.6% | 90.37% | 0.82× | 0.51× | 0.50× |
| | DemP-L2 | **71.52%** $\pm 0.09$ | 61.83% | 80.13% | 0.81× | 0.57× | 0.49× |
| | | **66.34%** $\pm 0.16$ | 74.1% | 89.93% | 0.61× | 0.42× | 0.34× |

Table 2: Results for pruning a ResNet-50 trained on ImageNet with **SGDM** at approximately 80% and 90% weight sparsity, formatted similarly to Table 1. DemP outperforms comparable structured pruning methods and is also compared with selected unstructured and restructured approaches, including recent DST methods like RigL, SRigL and Chase. While these methods achieve better sparsity-performance tradeoffs in terms of theoretical FLOP reduction, their sparsification is not fully GPU-amenable, leading to minimal training and inference speedup, as shown in the last column.

| | Method | Test accuracy | Neuron sparsity | Weight sparsity | Training time | Training FLOPs | Inference FLOPs | GPU Inf. FLOPs |
|---|---|---|---|---|---|---|---|---|
| | Dense | 75.62% ±0.28 | - | - | 1.0× | 1.0× (3.15e18) | 1.0× (8.2e9) | 1.0× (8.2e9) |
| Structured | SNAP | 26.05% ±0.17 | 36.90% | 81.40% | 0.54× | 0.33× | 0.33× | 0.33× |
| | | 24.84% ±0.11 | 56.07% | 90.10% | 0.47× | 0.25× | 0.25× | 0.25× |
| | CroPit-S | 25.93% ±0.06 | 36.90% | 81.40% | 0.50× | 0.31× | 0.31× | 0.31× |
| | | 25.44% ±0.05 | 53.20% | 89.80% | 0.49× | 0.26× | 0.26× | 0.26× |
| | EarlySNAP | 70.06% ±0.16 | 38.73% | 79.97% | 0.87× | 0.57× | 0.56× | 0.56× |
| | | 64.25% ±0.10 | 57.23% | 89.73% | 0.78× | 0.44× | 0.44× | 0.44× |
| | EarlyCroP-S | 71.45% ±0.29 | 41.13% | 79.77% | 0.86× | 0.61× | 0.61× | 0.61× |
| | | 67.04% ±0.18 | 58.77% | 90.07% | 0.80× | 0.48× | 0.48× | 0.48× |
| | DemP-Lasso($\gamma$) | **72.21%** ±0.19 | **53.50%** | 79.43% | 0.88× | 0.62× | 0.58× | 0.58× |
| | | **67.76%** ±0.12 | **68.51%** | 89.73% | 0.69× | 0.46× | 0.41× | 0.41× |
| | Dense[†] | 76.67% | - | - | - | - | - | |
| | Dense*¶§ | 76.8 ±0.09 % | - | - | - | 1.0× | 1.0× | 1.0× |
| Unstructured | Mag[†] | **75.53%** | - | 80% | - | - | - | 1.0× |
| | Sal[†] | 74.93% | - | 80% | - | - | - | 1.0× |
| | SET* | 72.9% ±0.39 | - | 80% | - | 0.23× | 0.23× | 1.0× |
| | | 69.6% ±0.23 | - | 90% | - | 0.10× | 0.10× | 1.0× |
| | RigL (ERK)* | 75.10% ±0.05 | - | 80% | - | 0.42× | 0.42× | 1.0× |
| | | 73.00% ±0.04 | - | 90% | - | 0.25× | 0.24× | 1.0× |
| Restructured | SRigL¶ | 75.01% | ≈2% | 80% | - | 0.36× | 0.41× | ≈1.0× |
| | | 72.71% | ≈5% | 90% | - | 0.24× | 0.24× | ≈1.0× |
| | Chase§ | 75.27% | 40% | 80% | - | 0.39× | 0.37× | 0.68× |
| | | **74.03%** | 40% | 90% | - | 0.26× | 0.23× | 0.67× |

[†]values obtained from Lee et al. (2023)

*values obtained from Evci et al. (2020)

¶values obtained from Lasby et al. (2024), neuron sparsity estimated from Fig. 3b

§values obtained from Yin et al. (2023)

a ResNet-50 on ImageNet (refer to Table 2). Additionally, DemP provides significant speedup when training on ImageNet with both optimizers, surpassing baselines yet again (Tables 1 and 2).

Furthermore, as shown in Fig. 15, these conclusions hold when a ResNet-18 is trained with Leaky ReLU activations instead of ReLU, highlighting DemP's adaptability across different activation functions.

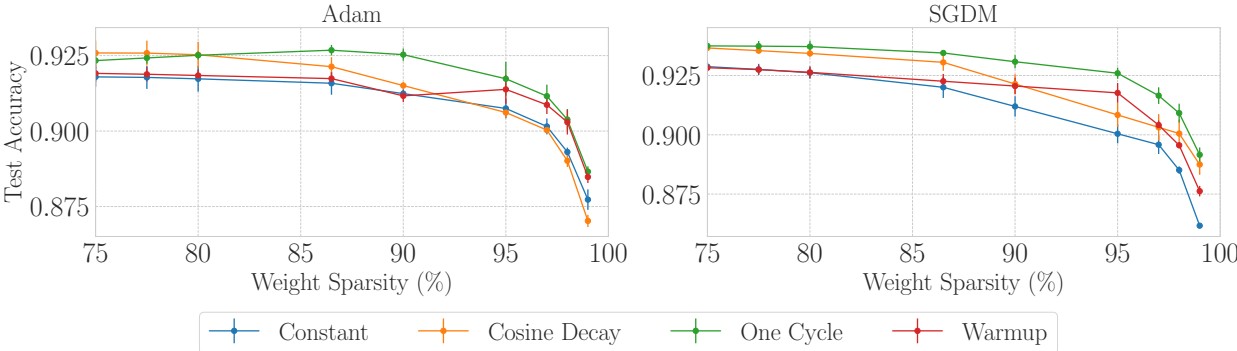

Figure 4: Impact of different schedules over the regularization parameter for DemP, concluding that a one-cycle scheduler is a good default choice. Experiments were performed with ResNet-18 on CIFAR-10, with Lasso($\gamma$) regularization, across three seeds. Higher sparsities are obtained by increasing the peak strength of the added scheduled regularization. **Left:** With Adam optimizer. **Right:** With SGDM optimizer.

## 4.2 Ablation

**Regularizers.** While the methodology works with both L2 and Lasso regularization, and either applied to the entire parameter space or specifically to the scale normalization parameters ($\gamma$), we empirically found that Lasso regularization of the scaling worked best for SGDM (Fig. 5). For Adam, L2 regularization of the scaling slightly outperforms Lasso regularization of the scaling (Fig. 5). As such, applying regularization solely to the scale parameters is beneficial to performance. For simplicity, we opted for Lasso regularization of the scale parameters in all subsequent experiments. Nevertheless, in the absence of normalization layers, traditional regularization methods remain viable for inducing sparsity in the model

**Dynamic Pruning.** To realize computational gain during training, we dynamically prune the NN at every $t$ steps (with a default of $t = 5000$). Dead neurons are removed almost as soon as they appear, preventing their revival. This strategy enables faster training with no significant change in performance as shown in Fig. 7. We note that this smooth gradual pruning process is compatible with our approach in part because there is no added cost for computing the pruning criterion.

**Regularization Schedule.** We test our hypotheses about regularization scheduling by comparing the one-cycle method with warmup, cosine decay, and constant schedules. Empirically, we confirm in Fig. 4 that using a one-cycle scheduler for the regularization parameter ($\lambda$) is a good strategy (Appendix F.4).

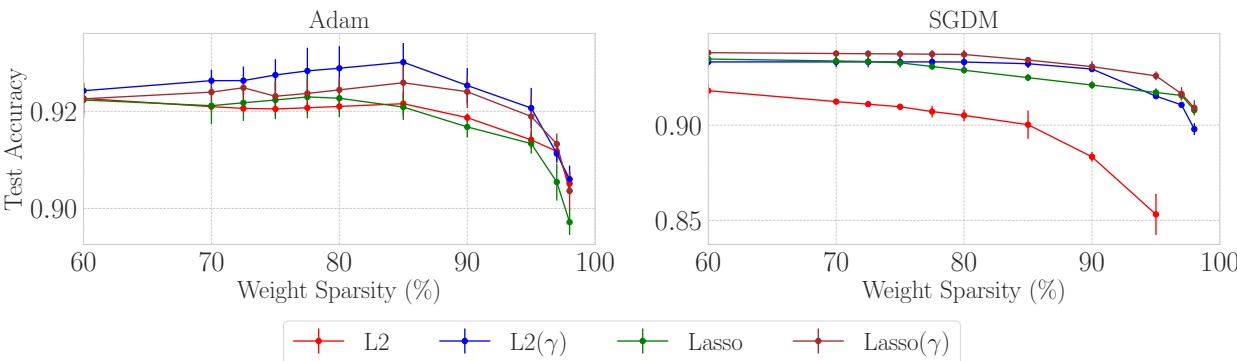

Figure 5: ResNet-18 networks trained on CIFAR-10 with different added regularization strategies, over three seeds. $\cdot(\gamma)$ denotes when regularization is only applied on the scale parameters of the normalization layer. Higher sparsities are obtained by increasing the peak strength of the added scheduled regularization. **Left:** With Adam, using L2($\gamma$) regularization slightly outperforms other strategies. **Right:** Using SGDM, the differences in performance become more pronounced, with Lasso regularization applied to scale parameters providing a favorable balance between sparsity and performance.

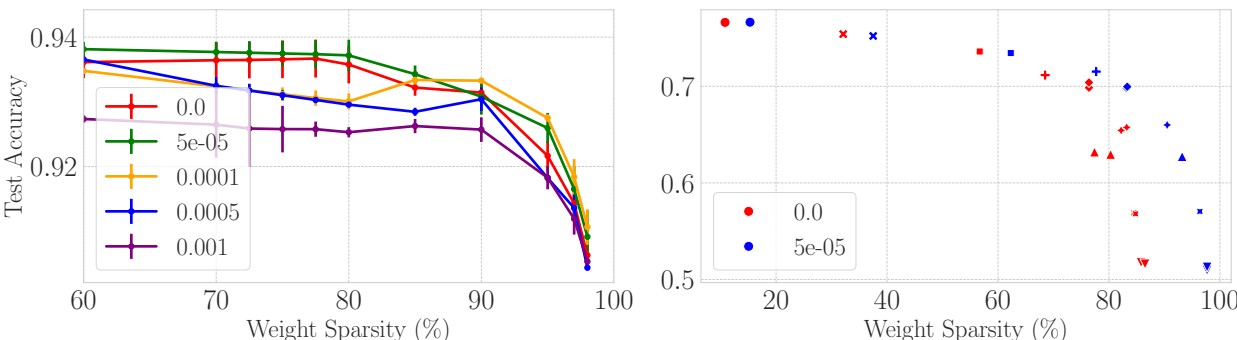

Figure 6: **Left:** ResNet-18 networks trained on CIFAR-10 with SGDM over three seeds. Adding a very small amount of Gaussian noise ($\sigma^2 = 5 \times 10^{-5}$) to the live neurons offers the best balance between performance and sparsity. **Right:** The addition of noise proves very impactful in increasing sparsity without affecting performance on a ResNet-50 trained on ImageNet. Each symbol represents a distinct peak regularization value for scale parameters, where each data point represents a distinct seed.

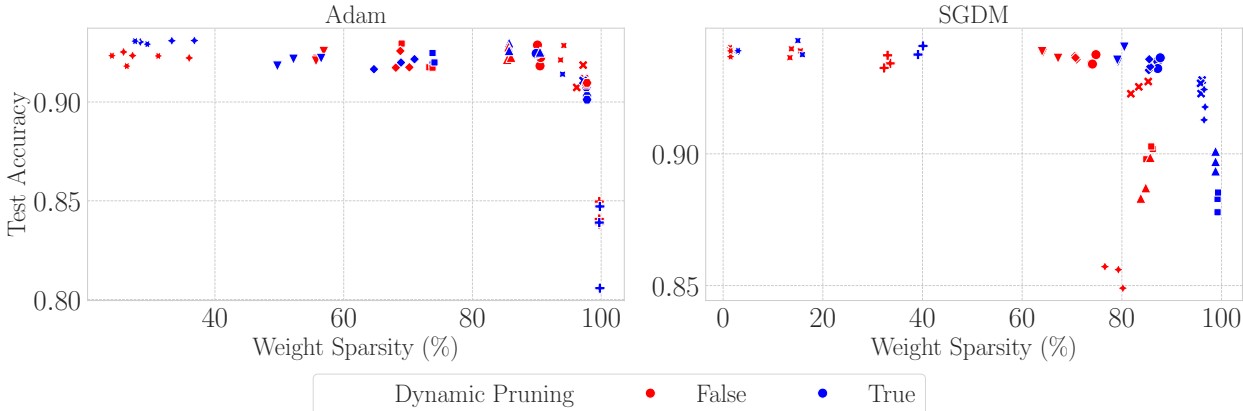

Figure 7: Measuring the impact of dynamic pruning on the accuracy-sparsity tradeoff. The symbols denote different regularization strengths. Experiments were performed with ResNet-18 on CIFAR-10 and Lasso($\gamma$) regularization (three seeds). **Left:** With Adam, there are minimal differences between the two strategies, indicating that dynamic pruning does not affect performance. **Right:** With SGDM, dynamic pruning further improves sparsity without degrading performance, while also providing training speedup.

**Added noise.** Adding asymmetric noise is particularly beneficial to further increase sparsity (see Fig. 6). The noise is kept small, with a peak variance at $\sigma^2 = 5 \times 10^{-5}$ and following the same schedule as the added regularization for maximal effect. We believe it acts as the small additional push helping neurons that linger close to the inactive region boundary to cross it.

We conducted experiments to assess the necessity of using asymmetric noise in DemP (Fig. 13). We found that when all other aspects of DemP remained constant (including dynamic pruning), asymmetric noise was not strictly essential. Yet, we note that dynamic pruning mimics the effect of asymmetric noise: removal of dead neurons also removes the possibility of random revival. We decided to keep the added noise asymmetric due to significant differences observed in other settings (Fig. 9, middle) where symmetric noise reduces dead neuron accumulation considerably due to neuron revival.

**Additional ablations** on the death criterion relaxation and weight decay are discussed respectively in Appendix F.3 and Appendix F.5. We also discuss the choice of activation functions in Appendix F.7, showing that DemP offers similar performance when using Leaky ReLU activations (Fig. 15).

Table 3: Using DemP to prune the MLP layers of a ViT-B/16 model. Solely by injecting artificial noise ($\lambda = 0$), DemP removes up to 90% of the neurons. We report the neural sparsity across the MLP layers but provide the equivalent weight sparsity for the entire network.

| Method | Test accuracy | Neuron sparsity | Weight sparsity | Training time |
|---|---|---|---|---|
| Dense | 78.75% | - | - | 1.0× |
| DemP | 71.38% | 90.6% | 59.4% | 0.77× |
| DemP + ReLU | 77.62% | 43.7% | 29.6% | 0.93× |

### 4.3 Transformers

To assess the compatibility of DemP with transformer-based architectures, we conducted exploratory experiments on Vision Transformer (ViT) networks (Dosovitskiy et al., 2021). Our focus was on pruning the fully connected layers because they contain the majority of the parameters and use GELU activation functions. Notably, we observed a substantial number of neurons dying during regular training, which led us to explore DemP's performance without regularization ($\lambda = 0$), retaining only the added noise.

The exploratory results for a ViT-B/16 model trained on ImageNet are presented in Table 3. We found that 90% of the fully connected neurons were readily pruned using DemP. Additionally, replacing the GELU activations with ReLU (Mirzadeh et al., 2024) resulted in only a slight performance degradation while still allowing the pruning of 43% of the neurons. Further details are provided in Appendix G.2.

## 5 Conclusion

In this work, we have explored how various hyperparameter configurations such as the learning rate, batch size, regularization, architecture, and optimizer choices, collectively influence activation sparsity during neural network training. Leveraging this, we introduced Demon Pruning, a dynamic pruning method that controls the proliferation of saturated neurons during training through a combination of regularization and noise injection. Extensive empirical analysis on CIFAR-10 and ImageNet demonstrated superior accuracy-sparsity tradeoffs compared to strong structured dense-to-sparse training baselines.

The simplicity of our approach allows for versatile adaptation. Integrating our method with existing pruning techniques is straightforward. Specifically, since DemP acts on the training dynamics, it can be used in conjunction with other dynamical pruning methods (Evci et al., 2020; Lasby et al., 2024; Yin et al., 2023). We discuss limitations of our approach in Appendix H.

**Broader Impacts Statement.** Structured pruning methods, even without specialized sparse computation primitives (Elsen et al., 2020; Gale et al., 2020), can efficiently leverage GPU hardware (Wen et al., 2016) compared to unstructured methods, which is crucial as deep learning models grow and environmental impacts escalate (Strubell et al., 2019; Lacoste et al., 2019; Henderson et al., 2020). Developing energy-efficient methods that can be widely adopted is essential.

However, while efforts to enhance the efficiency of deep learning training processes can reduce computational costs and energy requirements, they may inadvertently amplify concerns associated with the rapid advancement of AI. The swift progress in AI capabilities raises significant risks, from ethical dilemmas to information manipulation. By accelerating AI development and increasing its accessibility, research like ours may exacerbate ongoing issues.

### Acknowledgments

This research was enabled in part by compute resources provided by Mila, the Digital Research Alliance of Canada, and NVIDIA.

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

## A    Dead neurons in convolutional layers

In convolutional layers, ReLU is applied element-wise to the pre-activation feature map. We consider an individual neuron (channel) dead if all elements in the feature map after activation are 0. Formally, in this case, the definition from Section 2 becomes:

**Definition:** The $j$-th neuron/channel in the convolutional layer $\ell$ is *inactive* if it consistently outputs a feature map (post-activation) with elements summing to zero on the entire training set, i.e $\sum_{k,l} F_{jkl}^\ell = 0$. A neuron/channel that becomes and remains inactive during training is considered as *dead*.

## B    Biased Random Walk Model

This section aims to illustrate the role of asymmetric noise in neuron saturation through simple theoretical models, formalizing the intuition presented in Section 2.

**Setup.** We consider a network with parameter $\boldsymbol{w}$ trained to minimize the loss $L(\boldsymbol{w}) = \frac{1}{n} \sum_{i=1}^n \ell_i(\boldsymbol{w})$, where $\ell_i(\boldsymbol{w})$ is the loss function on sample $i$, using stochastic gradient descent (SGD) based methods. At each iteration, this requires an estimate of the loss gradient $g(\boldsymbol{w}) := \nabla L(\boldsymbol{w})$, obtained by computing the mean gradient on a random minibatch $b \subset \{1 \cdots n\}$. For simple SGD with learning rate $\eta$, the update rule takes the form

$$\boldsymbol{w}_{t+1} = \boldsymbol{w}_t - \eta \hat{g}(\boldsymbol{w}_t, b_t), \quad \hat{g}(\boldsymbol{w}, b) := \frac{1}{|b|} \sum_{i \in b} \nabla \ell_i(\boldsymbol{w}). \tag{2}$$

To formalize the intuition from section 2.1, we follow a standard line of work (Cheng et al., 2020) taking the view of SGD in Eq. 2 as a biased random walk (Anderson, 1998), described by the Langevin process,

$$\boldsymbol{w}_{t+1} = \boldsymbol{w}_t - \eta g(\boldsymbol{w}_t) + \sqrt{\eta}\, \hat{\boldsymbol{\xi}}(\boldsymbol{w}_t, b_t) \tag{3}$$

where the zero mean variable $\hat{\boldsymbol{\xi}}(\boldsymbol{w}, b) := \sqrt{\eta}\,(g(\boldsymbol{w}) - \hat{g}(\boldsymbol{w}, b))$ represents the gradient noise. In the limit of small learning rate, Eq. 3 is also well approximated (Cheng et al., 2020, Theorem 2) by the stochastic differential equation (SDE),

$$d\boldsymbol{w}_t = -g(\boldsymbol{w}_t) + M(\boldsymbol{w}_t)d\boldsymbol{B}_t, \tag{4}$$

where $\boldsymbol{B}_t$ denotes a standard Brownian motion and $M(\boldsymbol{w}) := \sqrt{\mathbb{E}_b[\hat{\boldsymbol{\xi}}(\boldsymbol{w}, b)\hat{\boldsymbol{\xi}}(\boldsymbol{w}, b)^\top]}$.

In SGD, a crucial characteristic of the gradient noise is its *multiplicative* nature, meaning it depends on the parameter. Systems with multiplicative noise exhibit a well-known property where regions with lower noise magnitude tend to act as attractors (Oksendal, 2010). Intuitively, the noise propels the system away from regions where it has a higher impact, leading to a higher probability of staying in regions where it has a lower impact. Mathematically, this manifests as a tendency for the invariant distribution associated with SDE in Eq. 4 to have a higher probability density in regions of lower noise magnitude.

We illustrate this phenomenon using two (drastically) simplified versions of the dynamics described by Eq. 3 and 4:

### B.1    Absorbing random walk

We consider a one-dimensional absorbing random walk with a boundary at zero described by the SDE:

$$dw_t = \begin{cases} \sqrt{\eta}dB_t & \text{as long as } w_t > 0 \\ 0 & \text{otherwise} \end{cases} \tag{5}$$

It models a system subject to noise in an 'active' region $w > 0$, which gets stopped at 0—and remains there, hence 'dies'—once it hits 0. It can be thought of as a simplified description of a regime where the dynamics (Eq. 3) is dominated by noise, such as e.g., a neuron encoding features with very low correlation to the task.

The *survival probability* at time $t$ is the probability that the system is still active $t$, i.e $w_t > 0$. It is related to the distribution of the first hitting time at 0 of a standard Brownian motion, $P(T_0 > t)$, where $T_0 = \inf\{t \geq 0 : B_t = 0\}$. A well-known property of Brownian motion (Karatzas & Shreve, 2014) is $\lim_{t \to \infty} P(T_0 > t) = 0$, which shows that the system in Eq. 5 eventually dies with probability 1. More generally, the following proposition specifies the dependence on learning rate and initialization:

**Proposition B.1.** *Consider the system (5) initialized at $w_0 > 0$. The survival probability at time $t > 0$ is given by*

$$P(w_t > 0 \,|\, w_0) = \text{erf}\left(\frac{w_0}{\sqrt{2\eta t}}\right) := \sqrt{\frac{2}{\pi}} \int_0^{w_0/\sqrt{\eta t}} e^{\frac{-w^2}{2}} \, dw \tag{6}$$

Prop. B.1 (*i*) confirms that the system eventually dies almost surely, since for all $w_0 > 0$ the survival probability decays to 0 as $t \to +\infty$ ; (*ii*) implies that for any given finite horizon time $t$, the smaller the initialization, the more likely the system is to be dead at $t$, (*iii*) illustrates how a noisier environment (i.e higher diffusive coefficient $\eta$ representing the learning rate) accelerates this dying process.

*Proof.* This is a standard application of the reflection property of Brownian motions (e.g., Lawler, 2016). Let $\bar{w}_t$ be the solution of Eq. 5 with *no* boundary condition. For the same initial condition, $w_t$ and $\bar{w}_t$ have the same distribution as long as $w_t > 0$. Let $T_0 = \inf\{t \geq 0 : \bar{w}_t = 0\}$ be the first hitting time of $\bar{w}_t$ at 0. The survival probability of $w_t$ can be expressed in terms of the distribution function of $T_a$ as

$$P(w_t > 0 \,|\, w_0) = 1 - P(T_0 \leq t) \tag{7}$$

The reflection property states that $P(T_0 \leq t) = 2P(\bar{w}_t \leq 0)$. To show this, let us first use the law of total probability to decompose $P(T_0 \leq t)$ as

$$P(T_0 \leq t) = P(T_0 \leq t, \bar{w}_t \leq 0) + P(T_0 \leq t, \bar{w}_t > 0) \tag{8}$$

For the first term, we note that $\bar{w}_t \leq 0$ implies $T_0 \leq t$ with probability 1, so $P(T_0 \leq t, \bar{w}_t \leq 0) = P(\bar{w}_t \leq 0)$. For the second term, we note that by the strong Markov property, $\bar{w}_t := \bar{w}_t - \bar{w}_{T_0} : t \geq T_0$ is a (scaled) standard Brownian motion, whose distribution is symmetric about the origin: therefore $P(\bar{w}_t > 0 | T_0 \leq t) = P(\bar{w}_t < 0 | T_0 \leq t)$. Thus,

$$\begin{aligned}
P(T_0 \leq t, \bar{w}_t > 0) &= P(\bar{w}_t > 0 \,|\, T_0 \leq t)P(T_0 \leq t) \\
&= P(\bar{w}_t < 0 \,|\, T_0 \leq t)P(T_0 \leq t) \\
&= P(\bar{w}_t < 0, T_0 \leq t) \\
&= P(\bar{w}_t \leq 0, T_0 \leq t)
\end{aligned} \tag{9}$$

where we have used the fact that $P(\bar{w}_t = 0) = 0$. As before, this last term equals $P(\bar{w}_t \leq 0)$; and the reflection property is proved. Finally, $d\bar{w}_t = \sqrt{\eta}B_t$ is a scaled Brownian motion with initial value $w_0$, so $\bar{w}_t$ is normally distributed with mean $w_0$ and variance $\eta t$. Thus,

$$P(w_t > 0 \,|\, w_0) = 1 - 2P(\bar{w}_t \leq 0) = \sqrt{\frac{2}{\pi}} \int_0^{w_0/\sqrt{\eta t}} e^{-\frac{w^2}{2}} \, dw$$

$\square$

## B.2 Geometrical random walk

The second example illustrates the stabilizing effects of multiplicative noise for systems such as Eq. 3 near unstable critical points (Oksendal, 2010). Consider the SGD dynamics (Eq. 2) with a diagonal quadratic approximation of the loss around 0, i.e., we assume

$$\ell_i(\boldsymbol{w}) = \ell_i(0) + \frac{1}{2}\boldsymbol{w}^T H_i \boldsymbol{w}, \quad H_i = \text{Diag}(h_{i1}, \cdots h_{id}) \tag{10}$$

where $H_i$ is some sample-dependent diagonal matrix. In such a setting, we model the dynamics of each parameter by a geometrical random walk,

$$w_{t+1} = w_t - \eta(h + \zeta_t)w_t \tag{11}$$

where $h$ is one of the Hessian eigenvalues and $\zeta_t$ is a noise variable sampled from some zero mean distribution. The case of a negative eigenvalue ($h < 0$) is particularly interesting since it corresponds to an unstable direction (negative curvature) in the absence of noise. In what follows, to ensure the stability of the dynamics, we assume that the noise variable $\zeta$ is bounded and the learning rate is small enough to ensure that $\eta|\zeta| < 1$. We also assume the noise distribution is symmetric about 0.

**Lemma B.2.** *Let $\mu = \mathbb{E}[\log(1 - \eta(h + \zeta))]$. For all $\epsilon > 0$ and $\delta > 0$, there exists $t_0(\epsilon, \delta)$ such that for all $t \geq t_0$, with probability at least $1 - \delta$,*

$$e^{t(\mu - \epsilon)}|w_0| \leq |w_t| \leq e^{t(\mu + \epsilon)}|w_0| \tag{12}$$

*In particular, $w_t \to 0$ w.h.p whenever $\mu < 0$.*

*Proof.* This is a consequence of the law of large numbers applied to the mean $\bar{z}_t = \frac{1}{t}\sum_{j=0}^{t-1} z_j$ of *i.i.d* variables $z_j := \log(1 + \eta(h + \zeta_j))$: for all $\epsilon > 0$, $\lim_{t \to +\infty} P(|\bar{z}_t - \mu| < \epsilon) = 1$. This implies that w.h.p,

$$e^{t(\mu - \epsilon)} \leq e^{t\bar{z}_t} \leq e^{t(\mu + \epsilon)} \tag{13}$$

Now solving Eq. 11 yields

$$w_t = w_0 \prod_{j=1}^{t-1}(1 - \eta(h + \zeta_j)) = w_0 \prod_{j=1}^{t-1} e^{z_j} = w_0\, e^{t\bar{z}_t} \tag{14}$$

Combining with Eq. 13 proves Lemma B.2. □

**Lemma B.3.** *: There exists a range of values for the learning rate $\eta$ for which $\mu < 0$, making the direction stable w.h.p, despite having $h < 0$.*

*Proof.* This is a consequence of the inequality $\log(1 + x) \leq x - \frac{x^2}{2} + \frac{x^3}{3}$ for all $x > -1$. Applying this inequality to $x = -\eta(h + \zeta)$ and taking the average over $\zeta$ gives the upper bound

$$\mu \leq -\eta h - \frac{\eta^2}{2}(h^2 + \sigma^2) - \frac{\eta^3}{3}(h^3 + 3h\sigma^2) \tag{15}$$

where $\sigma^2 := E[\zeta^2]$ and we used $E[\zeta] = E[\zeta^3] = 0$ by symmety about 0. Now, the sign of this bound coincides with the sign of the degree 2 polynomials $P(\eta) := |h| - \frac{\eta}{2}(h^2 + \sigma^2) + \frac{\eta^2}{3}|h|(h^2 + 3\sigma^2)$. We note that $P(0) > 0$ and that for a small enough ratio $|h|/\sigma$, it has two positive roots bounding an interval on which $P(\eta) < 0$. One way to see this is to compute the minimum

$$\min_\eta P(\eta) = |h| - \frac{3}{16}\frac{(h^2 + \sigma^2)^2}{|h|(h^2 + 3\sigma^2)}, \tag{16}$$

and to observe that it goes to $-\infty$ as $|h| \to 0^+$ for fixed $\sigma^2$. □

## C  Few Dead Neurons Revive

While empirical observations have shown a gradual accumulation of dead neurons (Fig. 1), we also observed that neurons can revive (Appendix D.2). To better assess the potential impact of reviving neurons on performance, we measured the overlap ratio ($|X \cap Y|/\min(|X|, |Y|)$) between the historical set of dead neurons at previous iterations and the set of dead neurons at the current iteration. This methodology directly follows Sokar et al. (2023). The results in Fig. 8 show that most neurons (over 90%) inactive at any point during training end up dead at the final iteration. This—coupled with our results showing that dying neurons can be dynamically pruned during training without impacting performance (Appendix F.2)—strongly suggests that neurons becoming inactive at any point during training in ReLU networks do not contribute significantly to the final performance of the trained model.

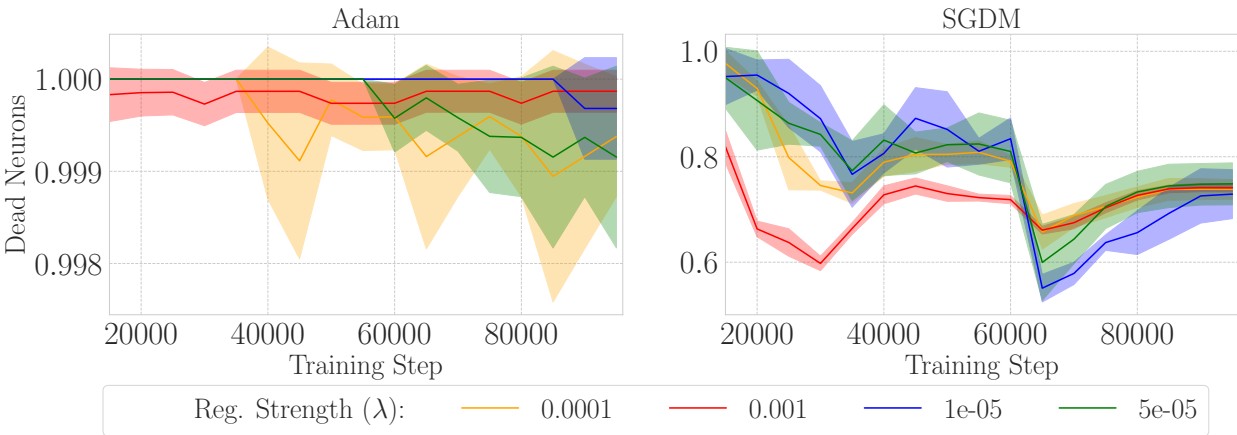

Figure 8: Overlap ratio of dead neurons during training, as measured across all layers of a ResNet-18 trained on CIFAR-10 at various maximal regularization strengths (Lasso($\gamma$)). Results are shown for training steps bigger than 15k when dead neurons become observable across all regularization strengths. The observation that most dying neurons remain dead justifies their early removal. **Left:** With Adam, we observe that almost all (over 99%) neurons dying never revive. **Right** The picture is more nuanced with SGDM, yet we find that the majority of dying neurons are still dead when training finishes ($\approx 75\%$). Neural revival mostly happens when the learning rate is decayed, followed right after by a phase where most of the revived neurons die again.

# D Hyperparameters Impact, Additional Results

## D.1 Noise, Learning Rate and Batch Size

To investigate the role of noise, we trained a 3-layer deep MLP (with layers of widths 100, 300, and 10 respectively) on a subset of 10 000 images of the MNIST dataset. To isolate the noise from a minibatch (of size 1) gradient ($\hat{g}(\boldsymbol{w}_j^t)$) we subtract from it the full gradient ($g(\boldsymbol{w}_j^t)$), taken over the entire training dataset. As such, the update of neuron $j$ at every time step is given by $\boldsymbol{w}_j^{t+1} = \boldsymbol{w}_j^t - \eta(\hat{g}(\boldsymbol{w}_j^t) - g(\boldsymbol{w}_j^t))$.

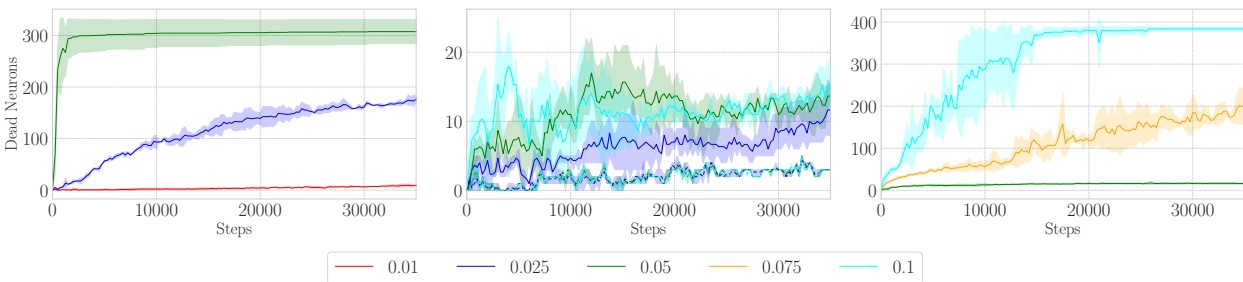

Figure 9: Evolution of the number of dead neurons for a 3-layer MLP (with layers of widths 100, 300, and 10 respectively) on a subset of MNIST. **Left:** The noisy part of the minibatch gradient is isolated and used exclusively to update the parameters. Noisy updates are *sufficient* to kill a subset of neurons following standard initialization. Because SGD gradient is 0 for dead neurons, there is an **asymmetry**: only live neurons are subject to noisy updates. **Center:** Gaussian noise is added to the parameters update, either asymmetrically (applied only to live neurons, plain lines) or symmetrically (dashed lines). Asymmetric noise is much more prone to dead neuron accumulation while symmetric Gaussian noise can revive neurons, contrary to SGD noise, leading to a much smaller accumulation. **Right** Standard SGD. Dead neurons accumulate quickly in noisy settings, but they plateau when the NN converges (leading to zero gradient). Results are averaged over three seeds.

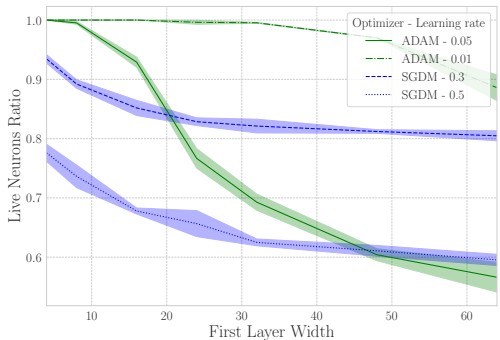
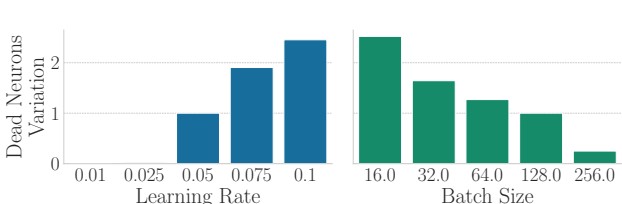

Figure 10: **Left**: An increased width leads to a higher ratio of neurons dying, independently of the optimizer. Measured on a ResNet-18 trained on CIFAR-10, without added regularization, across three seeds. We use the number of channels in the initial layer of the ResNet-18 to indicate the width, with 64 being the number of channels proposed by He et al. (2016) for the first convolution layer. **Right:** Varying the hyperparameters of a ResNet-18 (CIFAR-10) impacts the number of dead neurons. The bar heights indicate the multiplicative ratio of dead neurons compared to the base configuration (lr = 0.05, bs = 128 and $\lambda = 0$). The number of training steps was kept **constant** when varying the batch size for fair comparisons. Quantities are averaged over three random seeds.

This approach has the benefit of preserving the asymmetric noise structure of SGD updates (Wojtowytsch, 2023; Pillaud-Vivien, 2022), where dead units are not affected by noise but live ones are. Compared to applying solely symmetric Gaussian noise at every time step, we notice a much sharper accumulation of dead neurons. The details are shown in Fig. 9.

Diminishing the batch size or augmenting the learning rate also creates a noisier environment because both hyperparameters affect the noise covariance in SGD optimization (Keskar et al., 2017; Masters & Luschi, 2018; Goyal et al., 2017; He et al., 2019; Li et al., 2019). Furthermore, Smith et al. (2018) shows that learning rate decay can be replaced with batch size growth, emphasizing the relationship between the two quantities. Because of their impact on noise, we should expect those quantities to affect the dying ratio of neurons. We confirm empirically this hypothesis in Fig. 10.

## D.2    Training Time

The relation with training time, asserting that the probability of a neuron dying increases as training progresses (Prop. B.1) doesn't entirely align with practical applications. Modern overparameterized architectures often can memorize the entire training dataset, achieving zero loss in the process. Given that the gradient signal is proportional to the loss, it would concurrently diminish to zero for all neurons, preventing any further death.

We observe a pattern consistent with this idea (Fig. 1), where the total count of dead neurons spikes sharply in early training to then fluctuate slightly before stabilizing. The fluctuations demonstrate that neurons *can indeed revive*. However, additional experiments with ReLU networks revealed that most reviving neurons die again later (Fig. 8) and that their dynamic elimination has negligible to no impact on performance (Fig. 7).

## D.3    Network Width

The widths of a neural network's layers also influence the ratio of live neurons (live neurons to total neurons in the network) post-training (see Fig. 10). Typically, this ratio increases with the width; however, the total number of live neurons continues to rise with increased width. This phenomenon is somewhat anticipated as incorporating more neurons with random initialization in any given layer can only amplify the training noise, especially in the initial phase. Moreover, since initialization functions usually adjust their standard deviation proportionally to the number of channels ($\sigma \propto \sqrt{\frac{1}{\text{fan\_in+fan\_out}}}$), widening the network places neurons closer to their inactive region right from the initialization. The connection between width and

dead neurons maintains its significance as neural network sizes are inclined to increase over time with the availability of more computational resources. If this trend persists, the accumulation of dead neurons could potentially become increasingly pervasive.

## E    Adam is a Neuron Killer

The greater impact of Adam over the dying ratio compared to momentum must be due to the second-moment term, which is the only significant difference with momentum. Recall that Adam update (Kingma & Ba, 2015) is given by:

$$m_t = \beta_1 \cdot m_{t-1} + (1 - \beta_1) \cdot g_t$$
$$v_t = \beta_2 \cdot v_{t-1} + (1 - \beta_2) \cdot g_t^2$$
$$\hat{m}_t = \frac{m_t}{1 - \beta_1^t}$$
$$\hat{v}_t = \frac{v_t}{1 - \beta_2^t}$$
$$\theta_{t+1} = \theta_t - \frac{\eta}{\sqrt{\hat{v}_t} + \epsilon} \cdot \hat{m}_t$$

Earlier, we hypothesized that the neurons ending up dead were the ones experiencing very small gradients, such that the noise dominated their update trajectories. If this is the case, $g_t^2$ (the squared gradient) would be very small for those neurons' parameters, eventually leading to a very small second-moment estimation $\hat{v}_t$. In such a scenario, $\epsilon$ would end up dominating $\sqrt{\hat{v}_t}$, effectively multiplying the learning rate by $\epsilon^{-1}$ ($\epsilon$ is typically set to $1 \times 10^{-8}$). Moreover, as the decay ($\beta_2 = 0.99$) of $\hat{v}_t$ is usually slower than the one of $\hat{m}_t$ ($\beta_1 = 0.9$), a few sudden noisier updates would be sufficient to make huge random steps.

It is worth noting that RL practitioners typically set epsilon to a higher value (Hessel et al., 2018), as it has empirically been found to perform better. Because of constant distribution shifts, rapid accumulation of dead neurons often occurs in RL tasks (Lyle et al., 2022; Sokar et al., 2023). Higher $\epsilon$ values should reduce the number of dead neurons induced by Adam optimizer, by making the effective learning rate ($\frac{\eta}{\epsilon}$) smaller when $\epsilon$ dominates $\sqrt{\hat{v}_t}$. We confirm this hypothesis in Fig. 11, confirming the soundness of picking higher $\epsilon$ to improve performance and stability in RL.

Also notable, HuggingFace Transformers library (Wolf et al., 2020) default $\epsilon$ Adam parameter to $1 \times 10^{-6}$, following RoBERTa example (Liu et al., 2019). Manipulating the $\epsilon$ parameter of AdaGrad was also observed to impact significantly a transformer performance model (Agarwal et al., 2020). Confirming that those heuristic choices are due to their impact on dead neuron accumulation would be quite interesting.

## F    Pruning Method Ablation

We validate and justify the heuristic choices made for our pruning method, summarized in Algorithm 1, via empirical observation exposed in this section. We used the same setup as before for a ResNet-18 trained on CIFAR-10.

### F.1    Regularizer choice

In our empirical analysis, we evaluated the effectiveness of various regularizers on the performance of ResNet-18 networks trained on CIFAR-10, using either Adam or SGDM as optimizers. Our findings (Fig. 5) suggest that focusing regularization exclusively on scale parameters yields a more favourable balance between sparsity and performance with both optimizers. While L2 regularization on scale parameters slightly enhances performance, the scenario changes with SGDM, where Lasso regularization on these parameters outperforms others by a wider margin. Consequently, for the sake of simplicity in our experiments, we have chosen to consistently apply Lasso regularization to scale parameters.

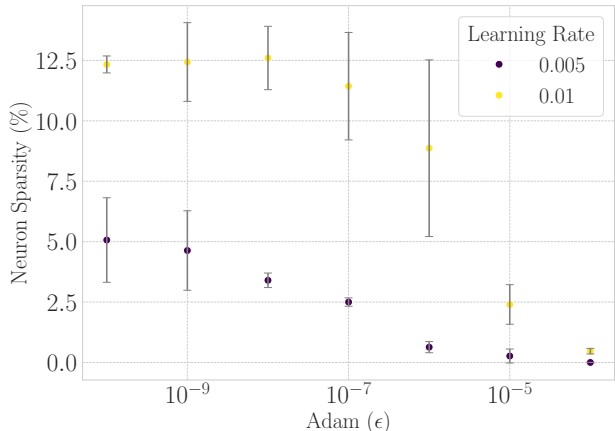

Figure 11: Sparsity decreases from $5.07\pm2.04\%$ to $0\%$ when varying Adam $\epsilon$ parameter from 1e-10 to 1e-4, or from $12.6\%\pm1.15\%$ to $0.47\%\pm0.01$ when the learning rate is increased, supporting our claim that the usage Adam optimizer impacts the accumulation of dead neurons through $\epsilon$ parameter. We note that this additional evidence complements the findings from Lyle et al. (2022), which reported reduced plasticity loss with increasing $\epsilon$. Experiments were performed with ResNet-18 on CIFAR-10 over three seeds, without DemP, using the same training recipe as in Appendix I.1.

### F.2 Dynamic Pruning

To verify the impact of dynamic pruning, we measured if there were any performance discrepancies when it was enabled or not. Across runs, we varied the regularization strength while measuring accuracy and sparsity. The results, in Fig. 7, show that enabling dynamic pruning does not affect the final performance. The very slight variations between runs fall well between the expected variance across different runs. This experiment reinforces the hypothesis that neurons that die and later revive during training do not contribute significantly to the learning process.

### F.3 Death Criterion Relaxation

The definition we choose for a dead neuron asks for it to be inactive to the entire dataset. In practice, we found that this criterion could be relaxed and defaulted to using 512 (2048 for ImageNet) examples from the training dataset to measure the death state (measuring across multiple minibatches when necessary). Fig. 12 shows that using this proxy for tracking dead units is sufficient. With this relaxation of the criterion, pruning interventions become highly efficient, incurring at most the computational cost of a few forward propagation batches. To measure if a minibatch could be used to measure the death state instead of the entire dataset, we tracked the number of dead neurons during training with both metrics in Fig. 12. We can see that both curves closely track each other. More importantly, they match at the end of the training, indicating that overall the same amount of neurons would be removed when performing the death check over the minibatch. Dynamic pruning was disabled for this experiment.

### F.4 Regularization Schedule

We also empirically tested different schedules over the regularization parameter in Fig. 4, trying to mitigate the impact of high regularization by decaying the parameter throughout the training after a warmup phase. We settled on using a one-cycle scheduler for the regularization strength because of slightly better performance in the higher sparsity level. However, we remark that all tested schedules over the regularization parameter remain sound with our method.

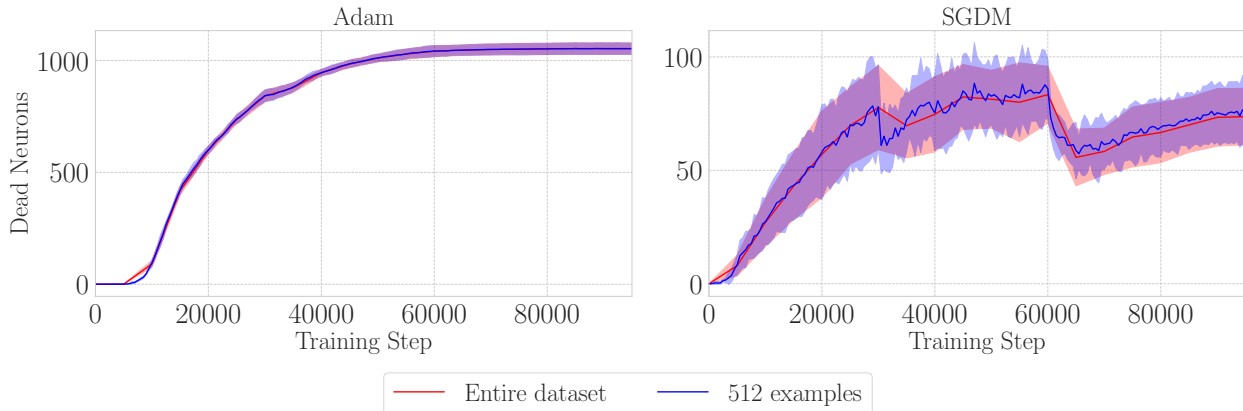

Figure 12: Instead of validating the death state of neurons against the entire training dataset, it proves sufficient to use a smaller dataset. The curves match throughout training, indicating that roughly the same number of neurons are removed with both strategies. Experiments were performed with ResNet-18 and Lasso($\gamma$) regularization on CIFAR-10 over three seeds. **Left:** With Adam. **Right:** With SGDM.

### F.5 Weight Decay

Our method defaults back to traditional regularization, with a term added directly to the loss, as opposed to the weight decay scheme proposed by Loshchilov & Hutter (2019). By doing so, the adaptive term in optimizers takes into account regularization, and neurons move more quickly toward the inactive region. From a pruning perspective, it achieves a higher sparsity than weight decay for the same regularization strength. Note that we are referring here to the added one-cycle regularization on scale parameters, which on top of potential constant weight decay applied as part of the original training recipe.

### F.6 Added Noise

We measured empirically the impact of adding artificial asymmetric noise. As expected, adding too much noise hurts performance. However, when noise variance is small enough, it doesn't affect performance while helping neurons to cross over in the inactive region, as detailed in Fig. 6. The effect is particularly significant when training a ResNet-50.

Furthermore, we assess that increasing noise instead of regularization is also a valid strategy for pruning, as showcased in Fig. 13. However, it leads to a worse tradeoff than when using increased regularization. Significantly, in this setting, with dynamic pruning enabled, we notice no major difference in performance between using asymmetric or symmetric noise.

### F.7 Activation function

Dead neurons are naturally defined with ReLU activation functions, for which neurons can completely deactivate. However, most activation functions, such as Leaky ReLU (Maas et al., 2013), also exhibit a "soft" saturated region. We postulate that neurons firing solely from the saturated region do not contribute much to the predictions and can be considered *almost dead*. We test this hypothesis by employing our method in a network with Leaky ReLU activations (Fig. 15), removing neurons with only negative activation across a large minibatch. Again, our method can outperform baselines with Adam and offers a similar performance with SGDM.

## G Additional Results

We report additional results in this section, including the pruning performance of DemP when pruning a VGG-16 (Fig. 14) trained with both Adam and SGDM and when pruning a ResNet-50 trained with SGDM

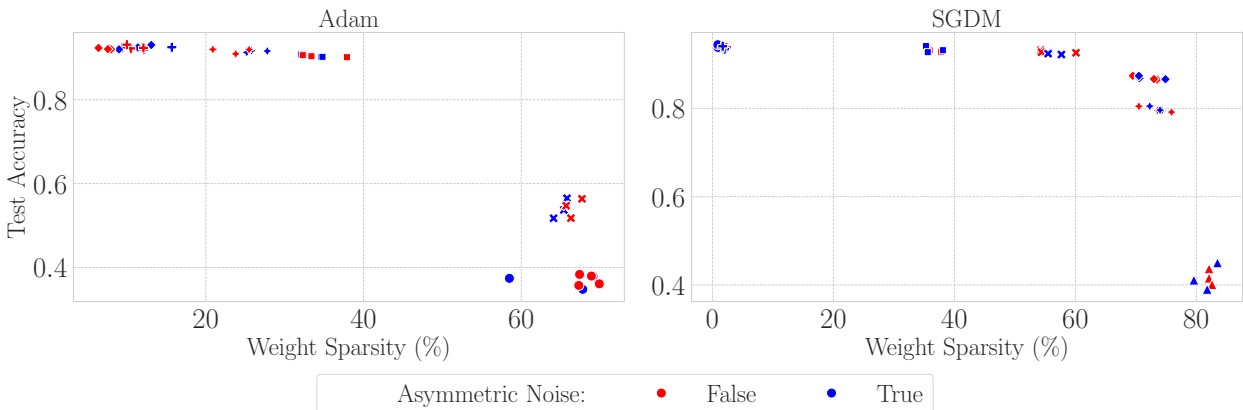

Figure 13: Weight sparsity for a ResNet-18 trained on CIFAR-10 while varying the variance of the added noise, *without adding any regularization.* The experiments were made with dynamic pruning enabled and showcased that using asymmetric noise over the live neurons only instead of symmetric noise across neurons does not impact the tradeoff. **Left:** With Adam, **Right:** With SGDM.

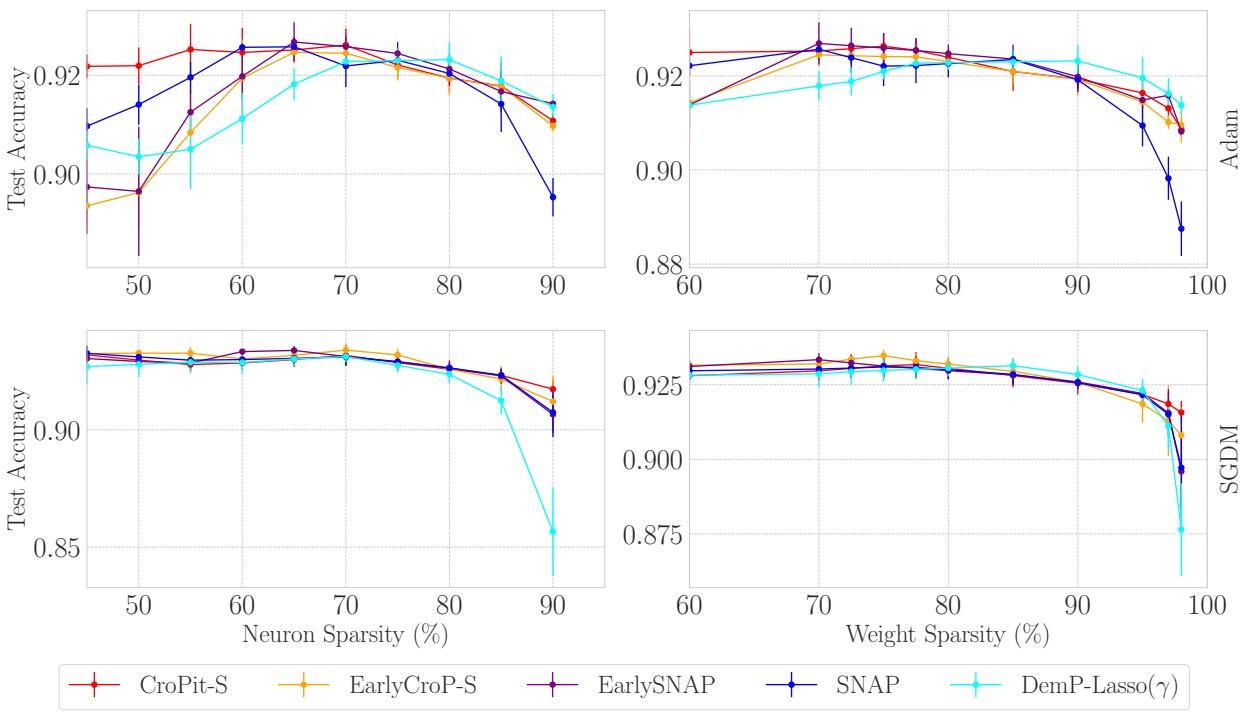

Figure 14: The results on VGG-16 networks trained with Adam (**Top**) and SGDM (**Bottom)** on CIFAR-10. With Adam at high sparsities, DemP can find subnetworks that better maintain performance. With SGDM, the performance instead decays quickly compared to baseline when reaching $\approx 95\%$ weight sparsity ($\approx 80\%$ neural sparsity). Higher sparsities with DemP are obtained by increasing the peak strength of the added scheduled regularization. **Left:** Neural sparsity, structured methods. **Right:** Weight sparsity, structured methods.

(Table 2). Results are similar, with DemP outperforming baselines with Adam at high sparsity. However, the performance of DemP with SGDM decay abruptly passed 95% weight sparsity, which may be due to improper tuning of the method when using a one-cycle learning scheduler to train for a small number of epochs (see Appendix I.2). The results with Leaky ReLU are in Fig. 15.

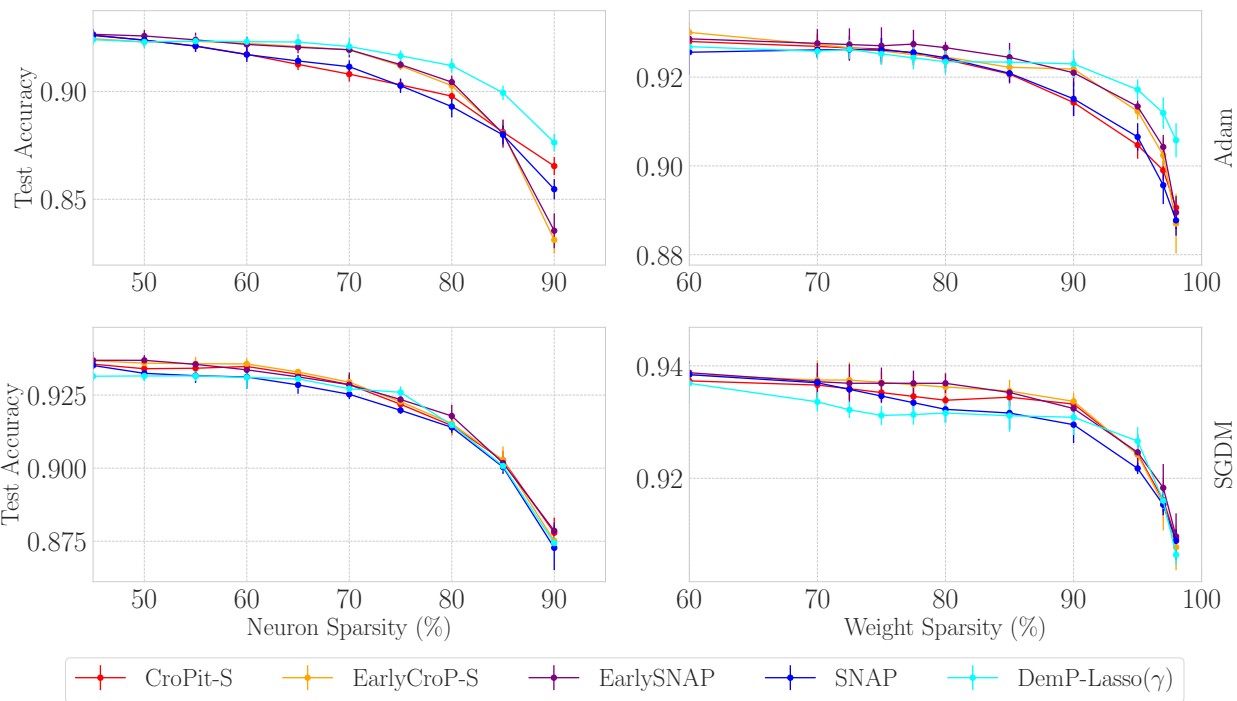

Figure 15: **Top:** ResNet-18 networks with *Leaky ReLU* trained on CIFAR 10 with Adam. DemP again outperforms the baseline structured pruning methods when using Adam. Higher sparsities with DemP are obtained by increasing the peak strength of the added scheduled regularization. **Bottom:** With SGDM, DemP performance is again similar to other methods. **Left:** Neural sparsity, structured methods. **Right:** Weight sparsity, structured methods.

We included results from Lee et al. (2023) and Evci et al. (2020) in Table 2 to better illustrate the tradeoff between structured and unstructured pruning methods. While unstructured methods currently offer more potential to maintain performance at higher parameter sparsity, structured methods offer direct speedup advantages.

## G.1 Comparison with Unstructured Methods

We employ the `JaxPruner` package (Lee et al., 2023) to illustrate further tradeoffs of our method against some unstructured methods. Our method is capable of achieving *similar* performance to unstructured ones for the ResNet-18 experiments (Fig. 16). The comparisons with the unstructured methods use their default configuration from JaxPruner, which was tuned for a ResNet-50. We expect their performance on smaller models to be improved by tuning the pruning distribution, the pruning schedule, and the pruning iterations scheme (Lee et al., 2023). However, for those not interested in expensive tuning, our method becomes an interesting default choice.

## G.2 Transformers Network

We explored the compatibility of DemP with transformer architectures, specifically the Vision Transformer (ViT-B/16) (Dosovitskiy et al., 2021) equipped with GeLU activation functions. Our focus was on pruning the MLP layers within the transformer blocks because these layers account for approximately 65% of the total parameters and contain activation functions, making them directly applicable to DemP.

This investigation led to two significant observations that, to the best of our knowledge, have not been previously reported: (a) even when training the "dense" version, 90% of the neurons within the MLP layers die during training, and (b) in the last MLP block, neurons tend to die quickly before gradually reviving. In

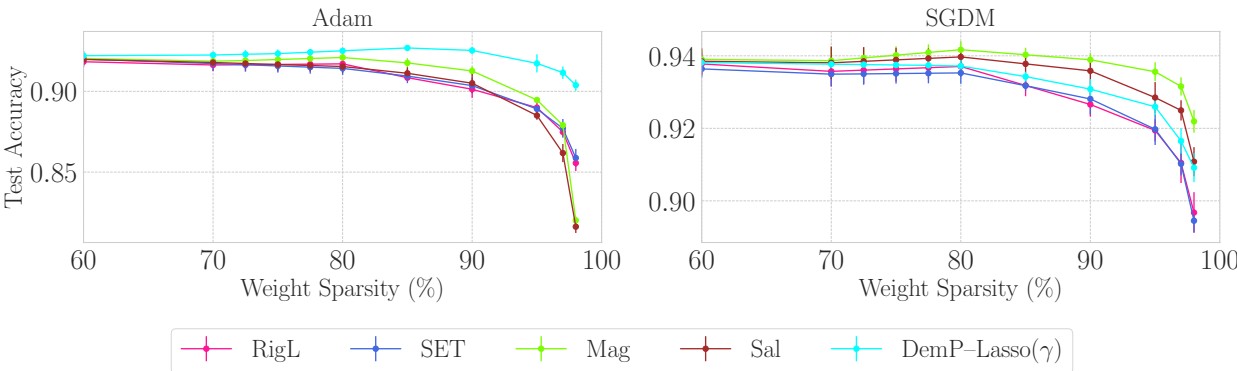

Figure 16: Comparison between DemP and unstructured pruning methods from Lee et al. (2023), performed on ResNet-18 trained on CIFAR-10 over three seeds. Higher sparsities with DemP are obtained by increasing the peak strength of the added scheduled regularization. **Left:** With Adam, DemP proves more effective than the unstructured methods left at their base configuration, even if it cannot achieve the same granularity in sparsity by being a structured method. **Right:** With SGDM, DemP performs similarly.

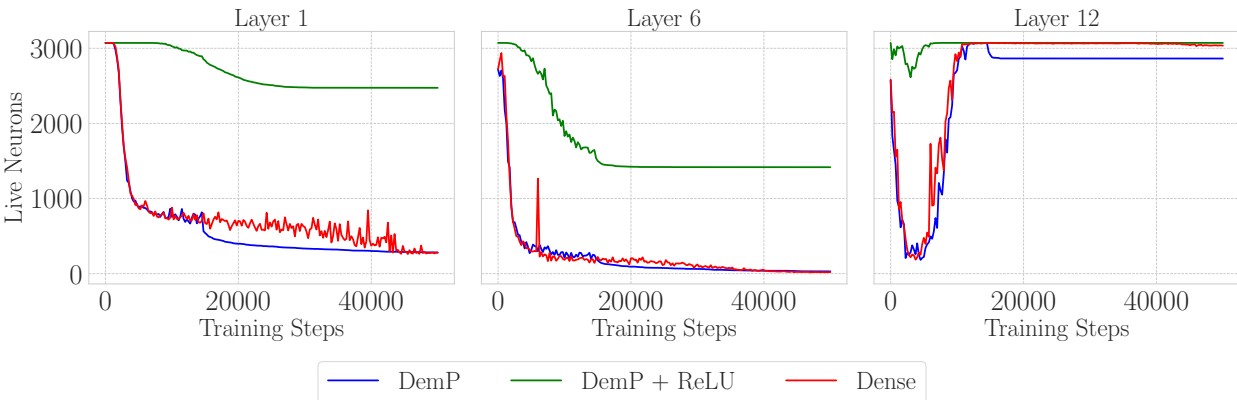

Figure 17: This figure illustrates the evolution of dead neurons during the training process for a 12-layer deep Vision Transformer (ViT-B/16). The rightmost plots depict the behavior in the last layer, where most neurons initially die but subsequently revive, exhibiting a qualitatively different pattern from the other layers. The leftmost and center plots show the typical behavior observed in the other layers, which is more consistent with what has been seen in other architectures.

other layers, neurons that die remain dead, consistent with our previous observations in other architectures. Those observations are reported in Fig. 17.

To accommodate this behavior, we adapted DemP so that pruning begins after neuron revival occurs in the last MLP layer. The exploratory results, reported in Table 3, are based on $\lambda = 0$ (no regularization), but with minimal additional noise applied. Inspired by Mirzadeh et al. (2024), we also included results where the GeLU activation functions were replaced with ReLU. Although these results are preliminary, they illustrate the potential of DemP for transformer-based architectures. A promising future direction would be to use DemP in conjunction with the approach proposed by Mirzadeh et al. (2024), where additional ReLU activations are incorporated into transformer architectures.

## H    Limitations of our approach

The primary limitation of our work is that, unlike most methods, the target sparsity level cannot be specified within the algorithm. Instead, DemP currently relies on a hyperparameter search over the regularization strength ($\lambda$) to achieve the desired sparsity. However, we are confident that our method can be adapted to this

paradigm by continuously increasing the regularization until the desired sparsity is achieved before terminating the process. This strategy is similar to the approach described by Wang et al. (2021). Additionally, while our analysis demonstrated the impact of batch size and learning rate on final sparsity, for practical reasons, we decided to rely entirely on regularization strength to control the sparsity level. This decision may constrain DemP to suboptimality.

In its current form, DemP removes entire channels from convolutional networks. Adopting a more fine-grained approach, such as removing individual filters, could potentially lead to better performance. Combining DemP with unstructured approaches could also result in a more competitive hybrid method.

Finally, DemP relies on activation functions with saturation regions to identify dead neurons. Consequently, DemP cannot be directly applied to prune layers that are not sandwiched between activation functions, such as the attention layers in transformer blocks. This is reflected in our experiments on ViT-B/16 networks (Section 4.3) where we limited DemP to pruning the fully-connected block within the network. Sparse training methods for Transformers remain underexplored and present unique challenges. A detailed study of DemP in this context, while valuable, is beyond the scope of this paper.

# I  Implementation details

## I.1  ResNet-18/ResNet-50

We mostly followed the training procedure of Evci et al. (2020) for the ResNet architectures.

**ResNet-18.** We train all networks for 250 epochs using a batch size of 128. The learning rate is initially set to 0.005 for Adam, to 0.1 for SGDM, and is thereafter divided by 5 every 77 epochs. While varying regularization is used with our method, it is on top of a constant weight decay (0.0005) used across all methods, including ours. Random crop and random horizontal flips are used for data augmentation. The training utilized a Nvidia RTX8000 GPU.

**ResNet-50.** We trained the ResNet-50 for 100 epochs, with a batch size of 256 instead of 4096. The initial learning rate is set to 0.005, before being decayed by a factor of 10 at epochs 30, 70, and 90. Label smoothing (0.1) and data augmentation (random resize to either $256 \times 256$ or $480 \times 480$, before randomly cropping to $224 \times 224$. Followed by random horizontal flip and input normalization) are also used. We again use Adam and SGDM, using constant weight decay (0.0001) for both. Training utilized a Nvidia A100 GPU.

## I.2  VGG-16

We followed a training procedure similar to Rachwan et al. (2022). We used Adam and SGDM with respectively a learning rate of 0.005 or 0.1 and a batch size of 256, with the One Cycle Learning Rate scheduler (Smith & Topin, 2018). The networks are trained for 80 epochs. CIFAR-10 images are normalized and resized to $64 \times 64$ before applying random crop and random horizontal flip for data augmentation. The training utilized a Nvidia RTX8000 GPU.

## I.3  Vision Transformer (ViT-B/16)

We followed the training routine of Lasby et al. (2024), that is we added data augmentations following the standard TorchVision (maintainers & contributors, 2016) ViT-B/16 training procedure for ImageNet. These augmentations include random cropping, resizing to 224x224 pixels, random horizontal flips, RandAugment (Cubuk et al., 2020), and normalization with typical RGB channel mean and standard deviation values. Additionally, we applied either random mixup (Zhang et al., 2018) or random cutmix (Yun et al., 2019), with alpha parameters of 0.2 and 1.0 respectively.

We sampled 16 mini-batch steps with 256 samples per mini-batch and accumulated gradients before applying the optimizer, resulting in an effective mini-batch size of 4096. The model was trained for 150 epochs using an AdamW (Loshchilov & Hutter, 2019) optimizer with weight decay, label smoothing, and $\beta_1$, $\beta_2$ coefficients of 0.3, 0.11, 0.9, and 0.999, respectively. We used cosine annealing with linear warmup for the learning rate scheduler, starting at 9.9e-5 and warming up to 0.003 by epoch 16. Gradients were clipped to a max L2 norm

of 1.0. Uniform sparsity was applied across all layers, with $\Delta T$ set to 100 to update connectivity every 100 mini-batch steps. The training utilized a Nvidia A100 GPU.

### I.4 Structured Methods

We closely reimplement in JAX (Bradbury et al., 2018) the structured methods from Rachwan et al. (2022), keeping all the hyperparameters specific to every method as is. The training hyperparameters are the same as specified in I.1 and I.2.

### I.5 Unstructured Methods

For the unstructured methods, we rely on Lee et al. (2023) implementations, using their method's configuration for pruning a ResNet-50 for all our experiments. The training hyperparameters are the same as specified in I.1 and I.2.

