# OpenReview forum: "Maxwell's Demon at Work: Efficient Pruning by Leveraging Saturation of Neurons"
_TMLR — Accepted by TMLR_

### Review · Reviewer_p6vp · 2024-12-18

**Summary Of Contributions:**

This paper investigates the effect of some neural network settings on the output magnitude of each neuron experimentally. From this finding, this paper introduces DemP, a method for inducing neurons to become non-functional during training and making the sparse feature space.
DemP achieves comparable or better accuracy on CIFAR-10 and ImageNet than traditional dynamic sparse training (DST) methods.

**Audience:**

Yes

**Claims And Evidence:**

Yes

**Requested Changes:**

Please see the weaknesses.

While this paper presents an interesting proposal with promising results, the writing needs improvement.

**Strengths And Weaknesses:**

**Strengths**
- It is an interesting idea to introduce noise in response to the neuron activity at an update time in order to manipulate the activity/inactivity of the neuron.
- Experiments was conducted on datasets and models of different scales. Also, this paper experimented with the Transformer architecture, which has recently been the subject of much discussion.

**Weaknesses**
- The research question needs to be clarified. I could not figure out what problem is being set in this work and what kind of answer DemP achieves to that problem.
- This paper introduces DemP as a pruning method, but reading this, it seems that DemP is not a pruning method. The main part of DemP is the proposal for the weight update part of the equation (1) and does not discuss the pruning saliency or something like that.
- Related to weaknesses 1 and 2 above, you must clarify what pruning methods DemP targets. After reading to its end, it is finally clear that structural pruning (channel-wise pruning?) and DST are DemP's targets. In my opinion, if you introduce equation (1), you can say that we use DemP, so you can also apply DemP to unstructured pruning, iterative pruning, and post-training pruning.
- Also, the main manuscript is hard to read because of frequent directions to the appendix and various sections to see important results. TMLR has no page limit, so the position of figures and some theoretical results should be modified to make it more readable.
- The comparison of methods needs to be more fair. This paper says that it adhered to the setting of the original paper for comparison, but this result does not tell me whether the good performance is due to one-cycled scheduling in Smith & Topin, 2018. Please also include the accuracy of previous studies on using one-cycled scheduling.
- Table 1 appears to be an important result, but it is difficult to see which aspect of DemP is superior to the conventional methods. It would be easier to convey them if the axes (e.g., neuron sparsity and accuracy) were selected and represented in a scatterplot.
- There is no comparison with other DST methods and DemP in the Transformer architecture. The result of this manuscript does not show the superiority of DemP over other DSTs in the Transformer, so I would like to see an evaluation of other DSTs in the Transformer. If the result of DemP with the Transformer is not good, please explain it in the limitation section.

**Minor typo**
- Section 2.2: similar to (Hu et al., 2016) -> similar to Hu et al., 2016
- Maybe the em dash command (---) is wrong. Also, please unify whether spaces are inserted before or after the em dash. (By the way, I prefer not to insert a space.)

---

> ### Author Response · Authors · 2024-12-20
> **Enhancing Clarity on DemP’s Goals, Comparisons, and Readability**
>
> We thank reviewer p6vp for their detailed and constructive feedback. Below, we address the key points raised and outline revisions (**highlighted in red** in the text) made to the manuscript.
>
> **Readability Improvements**
>
> We appreciate your observation on frequent references to the Appendix. To enhance clarity, we have:
> * Moved Algorithm 1 (pseudo-code of DemP) ant Table 2 (FLOPs reduction benefits) from the Appendix to the main text.
> * Reorganized Section 3 to improve the flow and provide a self contained narrative.
>
> We welcome further suggestion to enhance the paper’s readability.
>
> **Clarification of the  Research Question**.
>
> Our research addresses the challenge of structured pruning, specifically targeting the tradeoff between accuracy and sparsity. DemP proposes a novel sparse training method that:
>
> 1. Amplifies neuron death with minimal impact on predictions
> 2. Gradually removes dead neurons during training to achieve both training and inference speedup.
>
> **Positioning of DemP as a Pruning Method**
>
> We recognize the need for clearer framing in Section 3. DemP combines:
>
> - **Modified gradient updates** (Eq. 1) to promote neuron saturation and death.
> - **A pruning mechanism** to remove dead neurons iteratively during training.
>
> We revamped Section 3 to address your feedback. In particular, we added a subsection titled *Pruning Dead Neurons* to explicitly detail how regularization and noise amplify neuron death, which DemP leverages for pruning.
>
> **Comparison with one-cycle scheduling**
>
>  Thank you for highlighting this point. One-cycle scheduling is applied exclusively to DemP’s regularization parameter and is integral to our algorithm. While dynamic regularization has precedent [1], leveraging it specifically for pruning is a novel contribution. We clarified this in Section 3 and added comparisons where applicable.
>
> **Presentation of Results (Tables and Scatterplots)**
>
> **Table 1 and 2** are central to our results and align with prior work such as EarlyCrop [2], RigL [3], and Chase [5]. While scatterplots could provide additional insights, the tabular format was chosen to:
>
> 1. Maintain consistency with baselines.
> 2. Reduce computational overhead required for exhaustive scatterplots across sparsity levels.
>
> **Transformers and Baseline Comparisons**
>
> For our exploratory ViT-B/16 experiments in Section 4.3,  SRigL [1] is the only comparable work reporting results in this setting.  However, their pruning targets additional layers beyond MLPs, making direct comparison difficult. Chase [2] identifies sparse-friendly channels in ViTs but does not implement pruning, while methods like EarlyCrop [3] and Snap/EarlySnap [4] do not include results on Transformers.
>
> Sparse training methods for Transformers remain underexplored and present unique challenges. A detailed study of DemP in this context, while valuable, is beyond the scope of this paper. We clarified this limitation in the revised manuscript.
>
> ### References
>
> [1]: Yi Wang, Zhen-Peng Bian, Junhui Hou, Lap-Pui Chau: Convolutional Neural Networks With Dynamic Regularization. IEEE Trans. Neural Networks Learn. Syst. 32(5): 2299-2304 (2021)
>
> [2]: John Rachwan, Daniel Zügner, Bertrand Charpentier, et al. Winning the lottery ahead of time: Efficient early network pruning. In International Conference on Machine Learning, ICML 2022, PMLR, 2022
>
> [3]: Utku Evci, Trevor Gale, Jacob Menick, et al. Rigging the lottery: Making all tickets winners. In Proceedings of the 37th International Conference on Machine Learning, ICML 2020
>
> [4]: Mike Lasby, Anna Golubeva, Utku Evci, Mihai Nica, Yani Ioannou: Dynamic Sparse Training with Structured Sparsity. ICLR 2024
>
> [5]: Lu Yin, Gen Li, Meng Fang, Li Shen, Tianjin Huang, Zhangyang Wang, Vlado Menkovski, Xiaolong Ma, Mykola Pechenizkiy, Shiwei Liu: Dynamic Sparsity Is Channel-Level Sparsity Learner. NeurIPS 2023

---

> ### Comment · Reviewer_p6vp · 2025-01-02
>
> Thank you for taking the time to address my concerns.
>
> I checked the revised paper. It is easier to understand than the first draft. In particular, section 3 is easier to grasp DemP as a pruning method. I modified the claims and evidence score.
>
> The following are the additional requested changes for the revised paper:
>
> - Does the pseudocode and $\Delta R$ in Eq. (1) mean that $R$ shown in p6 is further differentiated by $w$? This $R$ doesn't contain a $w$ term, so I think $\Delta R $ is always $0$ in this case.
>
> - Sec. 2.1 and Sec. 3 refer to the theoretical result in App. B, so this result seems relatively important within the main manuscript. However, the main manuscript does not contain the section corresponding to the result. Please include informal (or formal) propositions in the main part so that readers can understand the theoretical results by reading only the main manuscript.
>
> - Similarly, I don't understand why Fig. 12, shown on p7, is in the appendix. If it is the important result to be cited in the important section of the proposed method, please introduce it in the main manuscript.
>
> - Doesn't it depend on the hardware implementation whether dynamic structural pruning affects training speed? Please show the speed results if you have already evaluated it. If you cannot evaluate it, the claim that DemP improves training speed is not a fact but a possibility, so please change your wording.
>
> - Typos:
>    - p1: only--- with -> only---with
>    - p3: Fig.  7 -> Fig. 7 (spacing between . and 7 is wrong)
>    - p4: in Fig.1, -> in Fig. 1
>    - p6: learning rate- -> learning rate---
>    - p7: Fig 12 -> Fig. 12
>    - p8: FLOPS -> FLOPs
>    - p9 Tab. 2: adn -> and; sparfication -> sparsification; minila -> minimal
>    - p22: in Fig. 9 -> in Fig. 9.
>
> Unfortunately, I found many typos. Please re-read the whole manuscript once.

---

> > ### Author Response · Authors · 2025-01-08
> >
> > Thank you for your acknowledgement of our rebuttal and the additional feedback.
> >
> > > Does the pseudocode and ΔR in Eq. (1) mean that R shown in p6 is further differentiated by w? This R doesn't contain > a w term, so I think ΔR is always 0 in this case.
> >
> > DemP applies regularization to a subset of the learnable parameters—the normalization scale parameters ($\gamma$).  We’ve improved our description of the regularization term—and in fact improved the formulation of the whole Section 3— in the new revision to make this crystal clear.
> >
> > > Sec. 2.1 and Sec. 3 refer to the theoretical result in App. B, so this result seems relatively important within the main manuscript. However, the main manuscript does not contain the section corresponding to the result. Please include informal (or formal) propositions in the main part so that readers can understand the theoretical results by reading only the main manuscript.
> > >
> >
> > Thank you for this suggestion. The theoretical results in Appendix B are included to support the intuition presented in Section 2.1 regarding the role of noise in neuron saturation. While insightful, these results do not claim novelty and are supplementary rather than central to our contributions. Thus, we believe they are appropriately placed in the Appendix.
> >
> > > Similarly, I don't understand why Fig. 12, shown on p7, is in the appendix. If it is the important result to be cited in the important section of the proposed method, please introduce it in the main manuscript.
> > >
> >
> > Fig. 12 demonstrates that the criterion for neuron inactivity (defined on p.4) can be relaxed from requiring dataset-wide evaluation to minibatch evaluation, enhancing DemP's computational practicality. This refinement does not affect sparsity-performance trade-offs, justifying its placement in the Appendix. In contrast, Fig. 7 highlights dynamic removal—a core feature of DemP—and is thus retained in the main manuscript.
> >
> > > Doesn't it depend on the hardware implementation whether dynamic structural pruning affects training speed? Please show the speed results if you have already evaluated it. If you cannot evaluate it, the claim that DemP improves training speed is not a fact but a possibility, so please change your wording.
> > >
> >
> > Training speed indeed depends on hardware implementation but is expected to improve on GPUs. We measured speed-up on an Nvidia A100, as shown in the training time column of Tables 1 and 2. These results, detailed further in Appendix I.1, substantiate the claim that DemP enhances training speed in practical applications.
> >
> > > Typos:
> > >
> >
> > Thank you for pointing these out. We have corrected the mentioned typos and conducted a thorough review of the entire manuscript using automated tools and manual checks. Please let us know if you notice any further issues.

---

> ### Comment · Reviewer_p6vp · 2025-01-08
>
> Thank you for your modification. I have checked your paper again.
>
> >Training speed indeed depends on hardware implementation but is expected to improve on GPUs. We measured speed-up on an Nvidia A100, as shown in the training time column of Tables 1 and 2. These results, detailed further in Appendix I.1, substantiate the claim that DemP enhances training speed in practical applications.
>
> I had overlooked this speed column of the table. Thank you for letting me know.
>
> This is minor point, but please correct the following. Maybe the reference is wrong.
>
> p3: further promote sparsity (Eq. ???)

---

> > ### Author Response · Authors · 2025-01-08
> >
> > > please correct the following. Maybe the reference is wrong.
> > > p3: further promote sparsity (Eq. ???)
> >
> > Done. Thank you!

---

### Review · Reviewer_Km7B · 2024-12-20

**Summary Of Contributions:**

The paper proposes DemP, which utilizes regularization and asymmetric noise injection in training to enforce sparse solutions by controlling the proliferation of dead neurons. The proposed approach achieves better accuracy-sparsity tradeoffs and faster training/inference compared to baselines such as SNAP, EarlySNAP, and EarlyCroPit.

**Audience:**

Yes

**Broader Impact Concerns:**

No broader impact concerns.

**Claims And Evidence:**

Yes

**Requested Changes:**

1. A more detailed discussion of regularization and asymmetric noise strengthen the paper.
2. Provide baselines performance in Table 2.

**Strengths And Weaknesses:**

Strengths:
1. The proposed method demonstrates superior performance compared to baselines like EarlySNAP and EarlyCroPit, achieving better accuracy-sparsity tradeoffs while reducing both training and inference costs.
2. The paper provides valuable insights into the phenomenon of neuron saturation.

Weaknesses:
1. The exact formulation of the regularization term $R$ is not explicitly provided, which makes it difficult to fully understand the proposed regularization strategy. Including this expression would improve the paper’s clarity. Moreover, the concept of asymmetric noise is not well-explained in Section 3.
2. Most of the experiments are limited to CNN with ReLU activations. And no baseline comparisons involving Vision Transformers are included.

---

> ### Author Response · Authors · 2024-12-20
> **Expanding on Regularization, Asymmetric Noise, and Transformer Experiments**
>
> We thank reviewer km7B for their constructive feedback. Below, we address the key points raised and outline changes (**highlighted in red** in the revised submission) made to the manuscript:
>
> **On regularization an asymmetric noise** :
>
> To improve clarity,  we revamped section 3 to include the exact formulation of the regularization term *R* and expanded the explanation of asymmetric noise.  We also added a new paragraph titled *Pruning Dead Neurons,* which clarifies how regularization and asymmetric noise are used to facilitate pruning. We welcome additional suggestions to further enhance readability.
>
> **On Experiments and Baseline Comparisons**
>
> Our experiments primarily focus on vision tasks (CIFAR-10/ImageNet) using VGG and ResNet architectures. To ensure broader coverage, we also included experiments with Leaky ReLU activations, as shown in Figure 15. Note that the original submission includes experiments with Leaky ReLU activations,  as shown in Fig. 15.
>
> For our exploratory ViT-B/16 experiments in Section 4.3,  to our knowledge, SRigL [1] is the only comparable work reporting results in this setting.  However, their pruning targets additional layers beyond MLPs, making direct comparison difficult. Chase [2] identifies sparse-friendly channels in ViTs but does not implement pruning, while methods like EarlyCrop [3] and Snap/EarlySnap [4] do not include structured pruning results on Transformers.
>
> Sparse training methods for Transformers remain underexplored and present unique challenges. A detailed study of DemP in this context, while valuable, is beyond the scope of this paper. We clarified this limitation in the revised manuscript.
>
> ### References
>
> [1]: Mike Lasby, Anna Golubeva, Utku Evci, Mihai Nica, Yani Ioannou: Dynamic Sparse Training with Structured Sparsity. ICLR 2024
>
> [2]: Lu Yin, Gen Li, Meng Fang, Li Shen, Tianjin Huang, Zhangyang Wang, Vlado Menkovski, Xiaolong Ma, Mykola Pechenizkiy, Shiwei Liu: Dynamic Sparsity Is Channel-Level Sparsity Learner. NeurIPS 2023
>
> [3]: John Rachwan, Daniel Zügner, Bertrand Charpentier, et al. Winning the lottery ahead of time: Efficient early network pruning. In International Conference on Machine Learning, ICML 2022, PMLR, 2022
>
> [4]: Stijn Verdenius, Maarten Stol, and Patrick Forré. Pruning via iterative ranking of sensitivity statistics. CoRR, abs/2006.00896, 2020.

---

### Review · Reviewer_LUGQ · 2024-12-21

**Summary Of Contributions:**

This paper researched the dying neurons phenomenon (i.e. inactive or saturated neurons) and proposed a simple yet effective structured pruning-based method Demon Pruning (DemP) to tackle this challenge. Based on the empirical observations of the dying neurons ratio during training, the authors thoroughly verify the influence of various factors, including regularization, noise, and optimizer. DemP is proposed upon these analyses and can actively promote neuron death, which can then be pruned dynamically during training. The dense-to-sparse pruning is implemented by applying decaying regularization and injecting noise early training. DemP is verified on ResNet/ViT across CIFAR-10 and ImageNet datasets, demonstrating the effectiveness and efficiency with a high compression ratio and significant speedup compared to baseline methods.

**Audience:**

Yes

**Broader Impact Concerns:**

All concerns have been sufficiently addressed in the Broader Impact Statement section

**Claims And Evidence:**

Yes

**Requested Changes:**

A recent pruning research HBGS[1] could be included in the discussion and comparison.

[1] A Greedy Hierarchical Approach to Whole-Network Filter-Pruning in CNNs, TMLR 2024

**Strengths And Weaknesses:**

Strengths:
- The presentation quality is good. The paper is easy to follow with informative figures.
- Analysis and observations on the impact factor of dying neurons are insightful and may inspire future research. It is also worth mentioning that the author analyzes the phenomenon both theoretically and empirically, which largely enhances the robustness and reliability of the research.
- Compared to structured and selected unstructured pruning baselines, DemP shows optimal accuracy and efficiency. Both the training FLOPs and inference FLOPs can be reduced.
- A comprehensive ablation study shows how the performance will change accordingly with different regularizers, regularization schedules, and pruning strategies.
An extended discussion on limitations is provided in the appendix, which should be encouraged.

Weaknesses:
- It's a bit overclaiming to say the DemP provides a "novel" perspective, as pruning by promoting dead neurons has been widely used in previous research [1][2].
- The early exploration of transformer-based methods is good. It will be better to include more tasks in addition to image classification, such as detection/segmentation or tasks in NLP (following [3]).

[1] Synaptic Stripping: How Pruning Can Bring Dead Neurons Back To Life, IJCNN 2023

[2] AP: Selective Activation for De-sparsifying Pruned Networks, TMLR 2023

[3] Winning the Lottery Ahead of Time: Efficient Early Network Pruning, ICML 2022

---

> ### Author Response · Authors · 2024-12-30
>
> We thank reviewer LUGQ for their constructive feedback. Below, we address the key points raised and outline revisions (highlighted in red in the text) made to the manuscript.
>
> **On novelty**:
>
> Thank you for highlighting the suggested references [1,2]. We acknowledge the extensive prior work on dead neurons. Methods like Synaptic Stripping [1] focus on mitigating the loss of plasticity caused by neuron death, while Activate-while-Pruning (AP) [2] aims to reduce the dynamic dead neuron rate, which can undermine standard pruning methods.
>
> DemP's novelty lies in fully leveraging neuron death as a mechanism to achieve high structured sparsity. Specifically, DemP amplifies neuron death during training through targeted regularization and asymmetric noise while iteratively removing dead neurons to refine the network. This combination sets DemP apart by utilizing neuron death as a feature rather than a limitation. We welcome suggestions to ensure this distinction is conveyed clearly and without ambiguity.
>
> **On benchmarking:**
>
> Thank you for introducing HBGS [3]. Our work focuses on dense-to-sparse pruning methods that adhere to a strict training budget while achieving practical reductions in both training and inference time. HBGS, as described, involves significant computational overhead to identify pruning targets and includes fine-tuning for 300 epochs post-pruning (Training Details, Section 4.1 [3]). In contrast, methods like DemP and EarlyCrop [4] enable real-time training acceleration without additional fine-tuning requirements.
>
> Our comparisons include Dynamic Sparse Training (DST) methods [5,6], which provide theoretical FLOP reductions during training, though these gains do not always translate to practical speedups.
>
> Structured sparse training for Transformers remains underexplored and presents unique challenges. For instance, DemP targets only MLP layers, while SRigL [5] includes additional layers but prunes unstructuredly before applying an intra-weight saliency criterion. Chase [6] identifies sparse-friendly channels in ViTs but does not implement pruning, while EarlyCrop [4] tested only its unstructured variant on Transformers.
>
> A detailed study of DemP's application to Transformers and additional tasks, such as detection, segmentation, or NLP, would be an exciting avenue for future work. However, we believe this is beyond the scope of the current paper.
>
> ## References
> [1] Synaptic Stripping: How Pruning Can Bring Dead Neurons Back To Life, IJCNN 2023
>
> [2] AP: Selective Activation for De-sparsifying Pruned Networks, TMLR 2023
>
> [3] A Greedy Hierarchical Approach to Whole-Network Filter-Pruning in CNNs, TMLR 2024
>
> [4]: John Rachwan, Daniel Zügner, Bertrand Charpentier, et al. Winning the lottery ahead of time: Efficient early network pruning. In International Conference on Machine Learning, ICML 2022, PMLR, 2022
>
> [5]: Mike Lasby, Anna Golubeva, Utku Evci, Mihai Nica, Yani Ioannou: Dynamic Sparse Training with Structured Sparsity. ICLR 2024
>
> [6]: Lu Yin, Gen Li, Meng Fang, Li Shen, Tianjin Huang, Zhangyang Wang, Vlado Menkovski, Xiaolong Ma, Mykola Pechenizkiy, Shiwei Liu: Dynamic Sparsity Is Channel-Level Sparsity Learner. NeurIPS 2023

---

### Decision · Action_Editor_awdz · 2025-01-26

**Recommendation:** Accept as is

**Comment:**

This paper proposed a dynamic pruning method by leverages dying neurons during the training of deep neural networks. The method explored various hyperparameter configurations, such as regularization and asymmetric noise injection, to enforce sparsity while preserving model accuracy. The results demonstrate improved accuracy-sparsity tradeoffs compared to existing structured dense-to-sparse training methods.

The paper is reasonally well-written and has generally received favorable reviews. The reasons for acceptance are:

1)	It demonstrates the viability and effectiveness of pruning based solely on neuron saturation.

2)	It offers insights into the neuron saturation behavior through the lens of sparsity and dynamic pruning.

3)	The results validate the effectiveness of the proposed method.

On the weakness side, the paper primarily focuses on ReLU-based activations, which may limit its application scope. Additionally, the writing can be improved by providing clearer comparisons with existing pruning methods to better highlights the advantages of proposed approach. Nevertheless, the strengths of the paper outweigh the weakness, therefore, it is recommended for acceptance.

**Audience:**

Yes. Both the insightful exploration of dying neurons and their applications in model pruning will be of interest to the deep learning community.

**Claims And Evidence:**

Yes